# Hyperbolic Hierarchical Alignment for Video-Based Visible-Infrared Person Re-Identification

**Shuang Li** [1 2]  **Changjiang Kuang** [1 2]  **Jiaxu Leng** [1 2]  **Mingpi Tan** [1 2]  **Zhanjie Wu** [1 2]  **Shuanglin Yan** [3]
**Xinbo Gao** [1 2]

## Abstract

Video-based visible-infrared person re-identification (VVI-ReID) aims to learn robust video-level representations under modality discrepancy. However, existing methods typically rely on Euclidean geometry, which is suboptimal for modeling the complex temporal dynamics within visible and infrared tracklets, as it inevitably distorts the intrinsic hierarchical structure inherent in diverse temporal variations (e.g., occlusion, pose). In this paper, we propose Hyperbolic Hierarchical Alignment (HHA), which unifies spatio-temporal modeling and cross-modality alignment on the Poincaré ball. HHA employs a Hyperbolic Hierarchical Spatio-Temporal Aggregator (HHSA) to organize time-varying cues into low-distortion hierarchical representations via Hyperbolic Geometry Interaction (HGI) and Dual-Geometry Fusion (DGF). Furthermore, we introduce Geometry-Aware Modality Alignment (GMA), which integrates Hyperbolic Modality Alignment (HMA) to couple modality centroids for geometric consistency and Hyperbolic Prototype Alignment (HPA) to anchor both modalities to shared identity prototypes for robust discrimination. Experiments on HITSZ-VCM and BUPTCampus demonstrate state-of-the-art performance. The code will be available at https://github.com/Visuang/HHA.

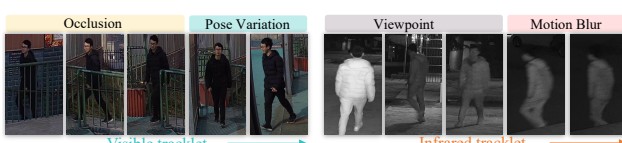

(a) Tracklets Exhibit Diverse Frame Cues

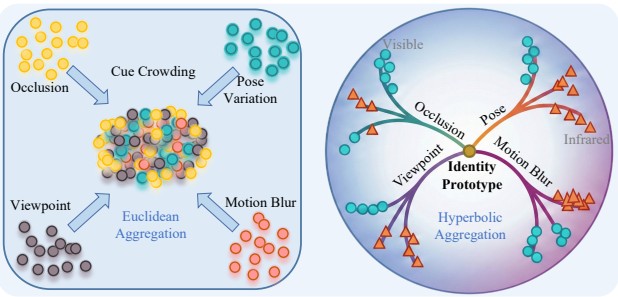

(b) Euclidean Cue Collapse vs. Hyperbolic Cue Aggregation

*Figure 1.* Tracklets in visible and infrared modalities exhibit diverse frame-wise cues (e.g., occlusion and pose variations), making spatio-temporal aggregation challenging. Euclidean aggregation entangles diverse cues, causing cue crowding and collapsed embeddings. Hyperbolic aggregation preserves cue-wise structure around an identity prototype for robust video-level representations.

## 1. Introduction

Person re-identification (ReID) (Ye et al., 2021b; Zhang et al., 2022; Li et al., 2023b; Su et al., 2025), aims to match pedestrian identities across disjoint cameras, serving as a fundamental technique for intelligent surveillance and public security (Yan et al., 2023; Leng et al., 2025; Su et al., 2026b;a; Mo et al., 2026). However, most existing ReID methods are designed for visible modality and static matching, which limits their practicality in real-world scenarios. In particular, visible cameras degrade under low illumination, and image-based ReID fails to exploit temporal cues from videos, motivating video-based visible-infrared person re-identification (VVI-ReID) (Lin et al., 2022; Du et al., 2023; Yang et al., 2025b) for matching pedestrians across visible and infrared video sequences.

Existing VVI-ReID methods primarily contend with two fundamental challenges. Some methods focus on alleviating the visible-infrared modality gap by learning modality-invariant embeddings in a shared feature space (Lin et al.,

[1]School of Computer Science and Technology, Chongqing University of Posts and Telecommunications, Chongqing, China [2]Chongqing Institute for Brain and Intelligence, Guangyang Bay Laboratory, Chongqing, China [3]College of Information Science and Technology, Nanjing Forestry University, China. Correspondence to: Jiaxu Leng <lengjx@cqupt.edu.cn>.

*Proceedings of the 43rd International Conference on Machine Learning*, Seoul, South Korea. PMLR 306, 2026. Copyright 2026 by the author(s).

2022; Li et al., 2023a; Zhou et al., 2025; Li et al., 2025b). The other emphasizes enhancing identity discrimination via spatio-temporal modeling, e.g., employing temporal modeling techniques such as pooling, attention, graph interactions, and transformers (Lin et al., 2022; Feng et al., 2024; Zhou et al., 2023). Despite significant progress, both approaches are typically realized in a standard Euclidean space, where temporal aggregation and modality alignment are coupled within a flat geometry. However, we argue that this flat geometry is suboptimal for modeling the intrinsic structure of video tracklets. As illustrated in Fig. 1 (a), a tracklet is not a single point but a hierarchical collection of cue-dependent realizations (e.g., frames under occlusion, pose variation, or viewpoint change) that naturally form a branching structure around an identity center. Euclidean geometry, restricted by polynomial volume growth, lacks the capacity to accommodate such expanding hierarchies without distortion. Consequently, Euclidean aggregation tends to compress these diverse cues into a crowded neighborhood, causing structural collapse and weakening identity discrimination, as shown in Fig. 1 (b). This issue is further exacerbated when modality discrepancy introduces additional heterogeneous variations.

In contrast to Euclidean geometry, hyperbolic geometry has continuous negative curvature, where the volume of the space expands exponentially with the radius rather than polynomially (Krioukov et al., 2010; Nickel & Kiela, 2017). This geometric property is theoretically optimal for embedding hierarchical data: a video tracklet can be structurally modeled as a tree, as illustrated in Fig. 1(b), with the global identity as the root and diverse temporal variations (e.g., occlusion, pose) as exponentially growing branches (Surís et al., 2021; Long et al., 2020). In Euclidean space, such branching structures are squeezed into a crowded embedding space, collapsing the hierarchical relations among diverse temporal variations (Sarkar, 2011; Sala et al., 2018); in contrast, hyperbolic space provides sufficient capacity to preserve these cue-wise hierarchies with lower distortion. Moreover, this geometry inherently facilitates Manifold Alignment. Unlike Euclidean approaches that often compromise local structure to minimize modality discrepancy, hyperbolic space allows the visible and infrared manifolds, which exhibit distinct cue-dependent hierarchies (Khrulkov et al., 2020), to be aligned non-linearly towards shared identity prototypes (Ghadimi Atigh et al., 2021). This enables geometry-consistent alignment that bridges the modality gap while preserving the intrinsic topological structure of each modality (Desai et al., 2023).

Motivated by the above analysis, we propose Hyperbolic Hierarchical Alignment (HHA), a geometry-aware framework that unifies spatio-temporal modeling and cross-modality alignment on the Poincaré ball. HHA exploits the exponential capacity of hyperbolic space to preserve the branching

structures of video tracklets while mitigating cross-modality distortion. The framework incorporates two core designs. First, Hyperbolic Hierarchical Spatio-Temporal Aggregator (HHSA) is designed to distill robust identity representations from sequential video frames. It integrates a Hyperbolic Geometry Interaction (HGI) module to progressively encode diverse cues into hierarchical memory, followed by Dual-Geometry Fusion (DGF) which fuses complementary Euclidean and hyperbolic attentions to capture discriminative identity cues. Second, we propose Geometry-Aware Modality Alignment (GMA) to explicitly bridge the visible-infrared gap. GMA synergizes Hyperbolic Modality Alignment (HMA), which couples identity-specific modality centroids to resolve manifold discrepancies, and Hyperbolic Prototype Alignment (HPA), which anchors these unified representations to shared identity prototypes to maximize identity discrimination. Together, HHA establishes an end-to-end hyperbolic framework for robust VVI-ReID.

Our main contributions are summarized as follows:

- We propose HHA, an end-to-end hyperbolic framework for VVI-ReID. To our knowledge, this is the first work to introduce hyperbolic geometry into VVI-ReID, alleviating the Euclidean geometric bottleneck.

- We design HHSA, which utilizes HGI to model temporal interactions while preserving branching structures, and employs DGF to excavate identity cues from the geometry-aware features via dual-geometry fusion.

- We introduce GMA, which synergizes HMA to couple modality centroids for cross-modality geometric consistency, and HPA to anchor representations to identity prototypes for robust discrimination.

- Extensive experiments on HITSZ-VCM and BUPT-Campus benchmarks demonstrate that HHA achieves state-of-the-art performance.

## 2. Related Work

### 2.1. Video-based Visible-Infrared Person Re-Identification

Video-based visible-infrared person re-identification (VVI-ReID) matches identities across heterogeneous modalities by learning discriminative video-level representations. Compared with image-based ReID (Li et al., 2025a; Teng et al., 2025; Yang et al., 2024), it benefits from temporal cues while facing greater modality discrepancy and sequence modeling challenges. Early studies focused on temporal aggregation for modality-invariant representation learning, such as MITML (Lin et al., 2022), while subsequent transformer-based methods (e.g., CST (Feng et al., 2024)

and IBAN (Li et al., 2023a)) modeled long-range cross-modal dependencies. Recent works improve robustness with stronger training strategies, e.g., AuxNet (Du et al., 2023) via auxiliary-sample curriculum and SAADG (Zhou et al., 2023) via style augmentation and disturbance defense. Meanwhile, model designs evolve toward multi-granularity spatio-temporal modeling (HD-GI (Zhou et al., 2025), X-ReID (Yu et al., 2026)) and CLIP-driven semantic integration (VLD (Li et al., 2025b), X-ReID (Yu et al., 2026)). Despite these efforts, unified geometry-aware modeling for spatio-temporal representation learning and cross-modality alignment remains underexplored.

### 2.2. Hyperbolic Representation Learning

Hyperbolic geometry provides a principled non-Euclidean embedding space with constant negative curvature, whose exponential volume growth is well suited for modeling hierarchical structures (Nickel & Kiela, 2017; Ganea et al., 2018; Chami et al., 2019; Surís et al., 2021). Poincaré embeddings (Nickel & Kiela, 2017) demonstrated that hyperbolic space can compactly encode latent hierarchies, with subsequent work extending hierarchical reasoning and representation learning (Ganea et al., 2018). Recent studies have applied hyperbolic embeddings beyond NLP and graphs, including uncertainty-aware and structured visual representations (Surís et al., 2021) and vision-language alignment via hierarchical semantics (Peng et al., 2025). In ReID, DS-VReID (Leng et al., 2025) (visible-only) validates Euclidean-hyperbolic complementarity; however, identity-centered hierarchical alignment of visible-infrared tracklets in hyperbolic space is still missing, motivating our unified hyperbolic VVI-ReID framework for spatio-temporal modeling and cross-modality alignment.

## 3. Methods

### 3.1. Hyperbolic Geometry and Framework Overview

We model video representations on the $n$-dimensional Poincaré ball $\mathbb{B}_c^n$ with constant negative curvature $-c$. Euclidean features are mapped to the manifold via $\exp_\mathbf{0}^c(\cdot)$ and projected back through $\log_\mathbf{0}^c(\cdot)$. We measure similarity using the hyperbolic distance $d_c(\cdot, \cdot)$. Hyperbolic geometry exhibits distances to grow exponentially with depth, enabling more efficient embedding and representation of hierarchical or tree-like structures (Nickel & Kiela, 2017; Ganea et al., 2018). Due to space limitations, detailed mathematical definitions are provided in Appendix A.

Based on this geometry, we propose Hyperbolic Hierarchical Alignment (HHA) for robust VVI-ReID learning (Fig. 2). HHA integrates a Hyperbolic Hierarchical Spatio-Temporal Aggregator (HHSA) into the visual encoder. Specifically, Hyperbolic Geometry Interaction (HGI) in-troduces a geometry-aware hyperbolic temporal interaction among intermediate frame tokens on $\mathbb{B}_c^n$ to preserve diverse discriminative cues, while Dual-Geometry Fusion (DGF) fuses Euclidean and hyperbolic attentions to aggregate these cues into identity-aware video representations. Finally, we employ Geometry-Aware Modality Alignment (GMA) to enforce cross-modality consistency and identity discrimination. GMA synergizes Hyperbolic Modality Alignment (HMA) to reduce cross-modality discrepancy and Hyperbolic Prototype Alignment (HPA) to anchor representations to stable identity prototypes, ensuring geometry-consistent alignment on the Poincaré ball.

### 3.2. Hyperbolic Hierarchical Spatio-Temporal Aggregator

Standard Euclidean temporal aggregation forces the intrinsic hierarchical structure of video tracklets containing diverse variations like occlusion and pose onto a flat embedding space, leading to high metric distortion and structural collapse. To address this geometric mismatch, we propose the Hyperbolic Hierarchical Spatio-Temporal Aggregator (HHSA), which incorporates hyperbolic geometry into the ViT backbone to preserve the underlying identity-variation structure. HHSA operates via two synergistic components: Hyperbolic Geometry Interaction (HGI) constructs hierarchical tokens through geometry-aware temporal interaction, and Dual-Geometry Fusion (DGF) fuses complementary Euclidean visual consistency and hyperbolic structural cues to excavate discriminative identity features.

**Hyperbolic Geometry Interaction.** To preserve diverse discriminative cues within the temporal hierarchy, HGI in-troduces a geometry-aware hyperbolic temporal interaction module on the Poincaré ball, enabling temporal interaction among intermediate CLS tokens. Let $\mathbf{f}_m^{t,l} \in \mathbb{R}^{B \times 1 \times D}$ de-note the CLS token of modality $m \in \{\text{vis}, \text{ir}\}$ extracted from the $t$-th frame at the $l$-th encoder block, where $B$ and $D$ represent the batch size and feature dimension, respectively. At selected layers, we map the CLS token of each frame to the Poincaré ball via:

$$\mathbf{z}_m^{t,l} = \exp_\mathbf{0}^c\left(\mathbf{f}_m^{t,l}\right), \tag{1}$$

where $\mathbf{z}_m^{t,l} \in \mathbb{B}_c^{B \times 1 \times D}$ is the hyperbolic embedding of $\mathbf{f}_m^{t,l}$ on the Poincaré ball. For each frame token, we then apply a hyperbolic linear layer $\phi_{\mathbf{W}^\mathbb{H}}(\mathbf{z}) = \mathbf{W}^\mathbb{H} \otimes_c \mathbf{z} \oplus_c \mathbf{b}^\mathbb{H}$, where $\mathbf{W}^\mathbb{H}$ and $\mathbf{b}^\mathbb{H}$ are learnable hyperbolic linear layer weight and bias, to generate the query, key, and value embeddings as:

$$\mathbf{q}_m^{t,l} = \phi_{\mathbf{W}_q^\mathbb{H}}(\mathbf{z}_m^{t,l}), \quad \mathbf{k}_m^{t,l} = \phi_{\mathbf{W}_k^\mathbb{H}}(\mathbf{z}_m^{t,l}), \quad \mathbf{v}_m^{t,l} = \phi_{\mathbf{W}_v^\mathbb{H}}(\mathbf{z}_m^{t,l}). \tag{2}$$

We compute the attention weight $\alpha_m^{t,\tau}$ from query frame $t$ to key frame $\tau$ by normalizing the pair-wise hyperbolic

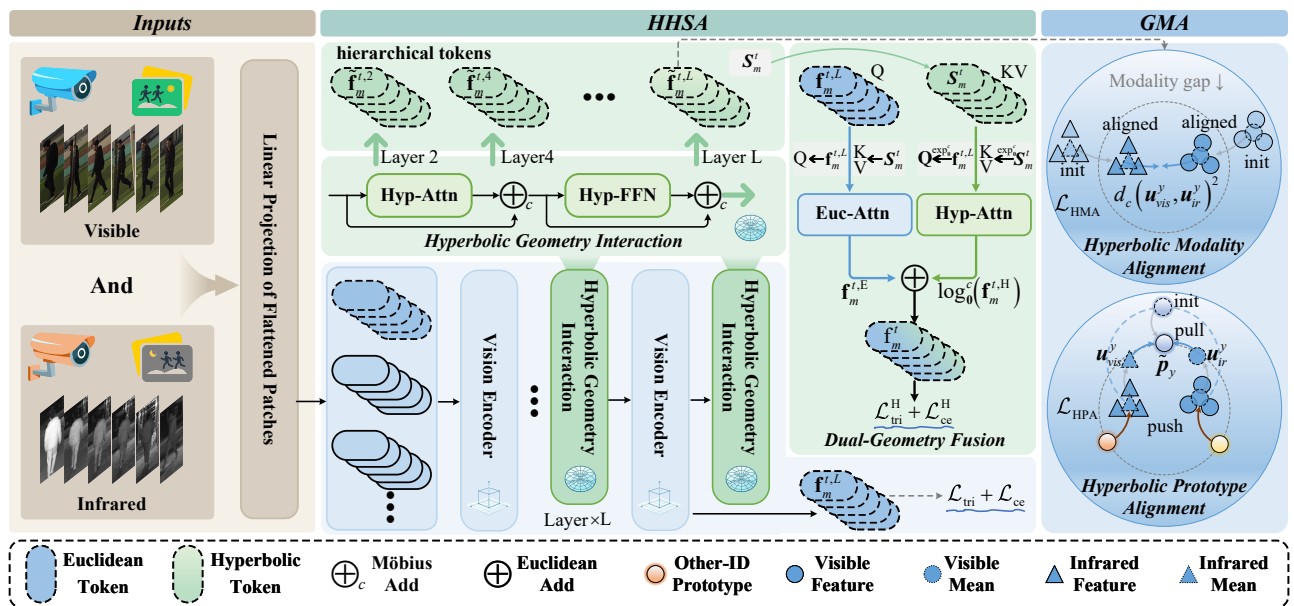

*Figure 2.* The proposed HHA framework learns robust video representations by introducing hyperbolic hierarchical modeling and alignment into a ViT-based encoder. Tracklet tokens are encoded by a ViT backbone with HHSA (HGI + DGF) for hierarchical spatio-temporal cue aggregation. HGI generates hierarchical tokens through hyperbolic temporal interactions, while DGF leverages dual-geometry attentions to excavate identity-relevant cues. The hyperbolic CLS embeddings are further constrained by HMA and HPA to reduce modality gap and enhance prototype-centered alignment on the Poincaré ball.

distances via the softmax function:

$$\alpha_m^{t,\tau} = \frac{\exp\left(-d_c(\mathbf{q}_m^{t,l}, \mathbf{k}_m^{\tau,l})^2\right)}{\sum_{\kappa=1}^{T} \exp\left(-d_c(\mathbf{q}_m^{t,l}, \mathbf{k}_m^{\kappa,l})^2\right)}. \quad (3)$$

Using these weights, we aggregate information across frames on the hyperbolic manifold via Möbius operations:

$$\bar{\mathbf{z}}_m^{t,l} = \mathbf{z}_m^{t,l} \oplus_c \phi_{\mathbf{W}^{\mathbb{H}}}\left(\bigoplus_{\tau=1}^{T}\left(\alpha_m^{t,\tau} \otimes_c \mathbf{v}_m^{\tau,l}\right)\right), \quad (4)$$

where $\oplus_c$ and $\otimes_c$ denote Möbius addition and scalar Möbius multiplication, respectively.

After cross-frame aggregation, we further refine the representation via a frame-wise hyperbolic feed-forward transformation with a Möbius residual update:

$$\hat{\mathbf{z}}_m^{t,l} = \bar{\mathbf{z}}_m^{t,l} \oplus_c \mathrm{FFN}_c^{\mathbb{H}}\left(\bar{\mathbf{z}}_m^{t,l}\right). \quad (5)$$

Here, $\mathrm{FFN}_c^{\mathbb{H}}(\cdot)$ denotes a hyperbolic feed-forward layer realized via $\log_{\mathbf{0}}^c(\cdot)$ and $\exp_{\mathbf{0}}^c(\cdot)$, preserving the hyperbolic geometry as follows:

$$\begin{aligned}
\mathbf{u} &= \phi_{\mathbf{W}_1^{\mathbb{H}}}\left(\bar{\mathbf{z}}_m^{t,l}\right), \\
\mathbf{v} &= \exp_{\mathbf{0}}^c(\sigma(\log_{\mathbf{0}}^{\mathbf{c}}(\mathbf{u}))), \\
\mathrm{FFN}_c^{\mathbb{H}}\left(\bar{\mathbf{z}}_m^{t,l}\right) &= \phi_{\mathbf{W}_2^{\mathbb{H}}}(\mathbf{v}).
\end{aligned} \quad (6)$$

where $\sigma(\cdot)$ is the GELU activation. After hyperbolic temporal interaction and refinement, we obtain a temporally

calibrated CLS token $\hat{\mathbf{z}}_m^{t,l}$. We then map the refined hyperbolic embedding back to the tangent space to obtain $\tilde{\mathbf{f}}_m^{t,l} \in \mathbb{R}^{B \times 1 \times D}$:

$$\tilde{\mathbf{f}}_m^{t,l} = \log_{\mathbf{0}}^c\left(\hat{\mathbf{z}}_m^{t,l}\right), \quad (7)$$

we then use the refined CLS token to replace the original CLS token of the $t$-th frame at layer $l$, allowing identity-aware temporal cues to guide subsequent Euclidean processing.

In addition, the representations produced at multiple HGI layers are preserved as hierarchical memory:

$$\mathbf{S}_m^t = \left\{\tilde{\mathbf{f}}_m^{t,\ell} \,\middle|\, \ell \in \mathcal{I}_{\mathrm{HGI}}\right\}, \quad (8)$$

here, $\tilde{\mathbf{f}}_m^{t,\ell}$ is a layer-specific summary token of the $t$-th frame, obtained by mapping the refined hyperbolic embedding back to the tangent space at the origin. The set $\mathbf{S}_m^t$ therefore forms a compact hierarchical memory in the tangent space, which is concatenated with patch tokens and propagated through subsequent ViT blocks.

**Dual-Geometry Fusion.** Although HGI encodes hierarchical spatio-temporal cues into the hierarchical memory $\mathbf{S}_m^t$, summarizing these cues into a compact identity representation remains challenging. We argue that a robust video representation must capture two complementary aspects of identity: static visual consistency (e.g., clothing texture) and dynamic structural variations (e.g., pose and visibility

changes under occlusion). Standard Euclidean attention excels at measuring visual similarity based on magnitude but struggles to interpret the hierarchical structures induced by HGI, whereas hyperbolic attention naturally models hierarchical entailment but may overlook flat visual correlations. Motivated by recent dual-space studies (Leng et al., 2025; 2024), we propose Dual-Geometry Fusion (DGF) to fully exploit these complementary inductive biases. Let $\mathbf{f}_m^{t,L}$ denote the final CLS token from the last ViT block. DGF employs $\mathbf{f}_m^{t,L}$ as an anchor query to retrieve identity-relevant cues from $\mathbf{S}_m^t$ via parallel attentions: Euclidean attention filters for appearance consistency, while hyperbolic attention excavates structural identity correlations.

In Euclidean space, DGF performs a CLS-guided attention readout over the hierarchical memory $\mathbf{S}_m^t$. Specifically, the final CLS token $\mathbf{f}_m^{t,L}$ serves as the query, while $\mathbf{S}_m^t$ provides the key-value set:

$$\mathbf{q}_m^{t,\mathrm{E}} = \mathbf{f}_m^{t,L}\mathbf{W}_Q^{\mathrm{E}}, \quad \mathbf{k}_m^{t,\mathrm{E}} = \mathbf{S}_m^t\mathbf{W}_K^{\mathrm{E}}, \quad \mathbf{v}_m^{t,\mathrm{E}} = \mathbf{S}_m^t\mathbf{W}_V^{\mathrm{E}},$$
(9)

where $\mathbf{W}_Q^{\mathrm{E}}, \mathbf{W}_K^{\mathrm{E}}, \mathbf{W}_V^{\mathrm{E}}$ are learnable projections. The Euclidean attention readout aggregates the hierarchical memory $\mathbf{S}_m^t$ into an identity-aware feature $\mathbf{f}_m^{t,\mathrm{E}}$ as:

$$\mathbf{f}_m^{t,\mathrm{E}} = \mathrm{softmax}\left(\frac{\mathbf{q}_m^{t,\mathrm{E}}(\mathbf{k}_m^{t,\mathrm{E}})^\top}{\sqrt{D}}\right)\mathbf{v}_m^{t,\mathrm{E}},$$
(10)

where $D$ is the feature dimension.

To further exploit the hierarchical relations stored in the hierarchical memory $\mathbf{S}_m^t$, we perform a geometry-specific attention readout in the hyperbolic space. Since both $\mathbf{f}_m^{t,L}$ and $\mathbf{S}_m^t$ are represented in the Euclidean space within the ViT backbone, we map them back to the Poincaré ball for geometry-consistent interaction:

$$\hat{\mathbf{f}}_m^t = \exp_{\mathbf{0}}^c(\mathbf{f}_m^{t,L}), \quad \hat{\mathbf{S}}_m^t = \exp_{\mathbf{0}}^c(\mathbf{S}_m^t).$$
(11)

For the $t$-th frame, we use $\hat{\mathbf{f}}_m^t$ as the query and $\hat{\mathbf{S}}_m^t$ as the key-value token memory. Using the hyperbolic linear layer $\phi_{\mathbf{W}^{\mathbb{H}}}(\cdot)$, we can obtain the hyperbolic query, key, and value embeddings:

$$\mathbf{q}_m^{t,\mathrm{H}} = \phi_{\mathbf{W}_Q^{\mathbb{H}}}(\hat{\mathbf{f}}_m^t), \mathbf{k}_m^{t,\mathrm{H}} = \phi_{\mathbf{W}_K^{\mathbb{H}}}(\hat{\mathbf{S}}_m^t), \mathbf{v}_m^{t,\mathrm{H}} = \phi_{\mathbf{W}_V^{\mathbb{H}}}(\hat{\mathbf{S}}_m^t).$$
(12)

Subsequently, we compute the attention weights based on the hyperbolic distance between the query and memory keys:

$$\alpha_{m,j}^{t,\mathrm{H}} = \frac{\exp\left(-d_c(\mathbf{q}_m^{t,\mathrm{H}}, \mathbf{k}_{m,j}^{t,\mathrm{H}})^2\right)}{\sum_{j'} \exp\left(-d_c(\mathbf{q}_m^{t,\mathrm{H}}, \mathbf{k}_{m,j'}^{t,\mathrm{H}})^2\right)},$$
(13)

where $j$ indexes the hierarchical token memory associated with frame $t$. The hyperbolic readout is then computed as

the Möbius weighted sum of the value embeddings:

$$\mathbf{f}_m^{t,\mathrm{H}} = \phi_{\mathbf{W}^{\mathbb{H}}}\left(\bigoplus_j (\alpha_{m,j}^{t,\mathrm{H}} \otimes_c \mathbf{v}_{m,j}^{t,\mathrm{H}})\right).$$
(14)

Finally, we fuse the Euclidean readout with the hyperbolic readout projected back to the tangent space:

$$\mathbf{f}_m^t = \mathbf{f}_m^{t,\mathrm{E}} + \log_{\mathbf{0}}^c(\mathbf{f}_m^{t,\mathrm{H}}),$$
(15)

where both terms lie in the Euclidean space and are combined by element-wise addition. We then obtain a tracklet-level representation by temporal average pooling, $\mathbf{f}_m^s = \frac{1}{T}\sum_{t=1}^T \mathbf{f}_m^t$, which serves as the final video representation for retrieval and is supervised by $\mathcal{L}_{tri}^H$ and $\mathcal{L}_{ce}^H$, where the superscript $H$ denotes the supervision on HHSA outputs.

### 3.3. Geometry-Aware Modality Alignment

While HHSA effectively aggregates spatio-temporal cues, it does not explicitly address the severe distribution shift between visible and infrared modalities. Empirically, without explicit geometric alignment, tracklets from different modalities tend to form disjoint regions on the manifold. Consequently, standard prototype learning becomes ill-posed, as a single prototype is forced to compromise between divergent modality manifolds, leading to ambiguous identity representations. To address this, we propose Geometry-Aware Modality Alignment (GMA), which jointly optimizes cross-modality consistency and identity discrimination on the Poincaré ball. GMA synergizes Hyperbolic Modality Alignment (HMA) to bridge disjoint modality sub-manifolds and Hyperbolic Prototype Alignment (HPA) to rigorously anchor representations to identity prototypes for robust discrimination.

**Hyperbolic Modality Alignment.** The primary challenge in VVI-ReID is that the modality-specific centroid of an identity in the visible domain often deviates significantly from its infrared counterpart due to sensor heterogeneity. Before learning global decision boundaries, it is crucial to eliminate this modality-specific offset. We first compute the modality-specific centroid (Fréchet mean) for each identity. Let $\mathcal{I}_m^y$ be the set of samples for identity $y$ in modality $m \in \{\mathrm{vis, ir}\}$. We approximate the centroid $\boldsymbol{\mu}_m^y$ by averaging in the tangent space and projecting back to the manifold:

$$\boldsymbol{\mu}_m^y = \exp_{\mathbf{0}}^c\left(\frac{1}{|\mathcal{I}_m^y|}\sum_{i \in \mathcal{I}_m^y} \log_{\mathbf{0}}^c(\mathbf{z}_{m,i}^s)\right),$$
(16)

where $\mathbf{z}_{m,i}^s$ is the tracklet embedding. HMA explicitly minimizes the hyperbolic distance between the centroids of the two modalities:

$$\mathcal{L}_{\mathrm{HMA}} = \sum_{y=1}^K d_c(\boldsymbol{\mu}_{\mathrm{vis}}^y, \boldsymbol{\mu}_{\mathrm{ir}}^y)^2,$$
(17)

*Table 1.* Values of mAP and CMC (%) obtained by our proposed method and the state-of-the-art Re-ID methods on HITSZ-VCM. "R@1", "R@5" and "R@10" denote Rank-1, Rank-5 and Rank-10, respectively.

| Methods | Reference | Seq_Len | Infrared-to-Visible | | | | Visible-to-Infrared | | | |
|---|---|---|---|---|---|---|---|---|---|---|
| | | | R@1 | R@5 | R@10 | mAP | R@1 | R@5 | R@10 | mAP |
| Lba(Park et al., 2021) | ICCV'21 | 6 | 46.4 | 65.3 | 72.2 | 30.7 | 49.3 | 69.3 | 75.9 | 32.4 |
| MPANet(Wu et al., 2021) | CVPR'21 | 6 | 46.5 | 63.1 | 70.5 | 35.3 | 50.3 | 67.3 | 73.6 | 37.8 |
| VSD(Tian et al., 2021) | CVPR'21 | 6 | 54.5 | 70.0 | 76.3 | 41.2 | 57.5 | 73.7 | 79.4 | 43.5 |
| CAJ(Ye et al., 2021a) | ICCV'21 | 6 | 56.6 | 73.5 | 79.5 | 41.5 | 60.1 | 74.6 | 79.9 | 42.8 |
| MITML(Lin et al., 2022) | CVPR'22 | 6 | 63.7 | 76.9 | 81.7 | 45.3 | 64.5 | 79.0 | 83.0 | 47.7 |
| SEFL(Feng et al., 2023) | CVPR'23 | 6 | 67.7 | 80.3 | 84.7 | 52.3 | 70.2 | 82.2 | 86.1 | 52.5 |
| CLIP-ReID (Li et al., 2023c) | AAAI'23 | 6 | 58.4 | 73.2 | 79.8 | 45.3 | 60.4 | 76.9 | 83.7 | 43.5 |
| IBAN(Li et al., 2023a) | TCSVT'23 | 6 | 65.0 | 78.3 | 83.0 | 48.8 | 69.6 | 81.5 | 85.4 | 51.0 |
| SAADG(Zhou et al., 2023) | ACM MM'23 | 6 | 69.2 | 80.6 | 85.0 | 53.8 | 73.1 | 83.5 | 86.9 | 56.1 |
| AuxNet(Du et al., 2023) | TIFS'23 | 6 | 51.1 | - | - | 46.0 | 54.6 | - | - | 48.7 |
| TF-CLIP (Yu et al., 2024) | AAAI'24 | 6 | 62.3 | 76.2 | 81.6 | 47.5 | 62.2 | 79.6 | 85.5 | 45.5 |
| CST(Feng et al., 2024) | TMM'24 | 6 | 69.4 | 81.1 | 85.8 | 51.2 | 72.6 | 83.4 | 86.7 | 53.0 |
| FA-Net(Yang et al., 2025a) | TIP'25 | 6 | 68.1 | 79.7 | 84.0 | 51.2 | 70.0 | 82.1 | 86.0 | 52.5 |
| STHF(Tao et al., 2026) | TCSVT'26 | 6 | 70.4 | 81.6 | 86.2 | 56.2 | 73.5 | 83.7 | 87.0 | 58.6 |
| VLD(Li et al., 2025b) | TIFS'25 | 6 | 74.3 | 85.0 | 88.4 | 60.2 | 74.6 | 86.4 | 90.0 | 58.6 |
| X-ReID(Yu et al., 2026) | AAAI'26 | 10 | 73.4 | 85.0 | - | 60.5 | 76.1 | 87.1 | - | 59.6 |
| **HHA(our)** | - | 6 | **76.0** | **85.6** | **89.3** | **63.2** | **77.5** | **88.3** | **91.4** | **60.8** |
| **HHA(our)** | - | 10 | **77.5** | **87.0** | **90.7** | **64.7** | **78.1** | **89.1** | **92.2** | **62.8** |

where $d_c(\cdot, \cdot)$ denotes the hyperbolic distance on the Poincaré ball. By minimizing the hyperbolic distance between $\boldsymbol{\mu}_{\text{vis}}^y$ and $\boldsymbol{\mu}_{\text{ir}}^y$, HMA enforces cross-modality feature consistency for each identity, providing a necessary geometric initialization for robust classification.

**Hyperbolic Prototype Alignment.** With the modality gap mitigated by HMA, we further introduce HPA to enforce discriminative representation learning. We define learnable hyperbolic prototypes $\mathcal{P} = \{\mathbf{p}_1, \ldots, \mathbf{p}_K\} \subset \mathbb{B}_c^n$ as the semantic anchors for each identity. Specifically, we parameterize these prototypes in the tangent space at the origin, $T_{\mathbf{0}}\mathbb{B}_c^n$, and project them onto the manifold via the exponential map $\mathbf{p}_k = \exp_{\mathbf{0}}^c(\tilde{\mathbf{p}}_k)$, where $\tilde{\mathbf{p}}_k \in \mathbb{R}^n$ are the learnable parameters. HPA anchors the aligned modality centroids to the corresponding shared prototypes. Unlike standard classifiers that only penalize individual samples, HPA imposes a structural constraint that pulls the unified modality centers towards the prototype:

$$\mathcal{L}_{\text{HPA}} = \mathcal{L}_{hce} + \lambda_p \sum_{y=1}^{K} \sum_{m \in \{\text{vis,ir}\}} d_c(\boldsymbol{\mu}_m^y, \mathbf{p}_y)^2, \quad (18)$$

where $\mathcal{L}_{hce}$ is the hyperbolic cross-entropy loss based on the probability $p(y|\mathbf{z}_m^s) \propto \exp(-d_c(\mathbf{z}_m^s, \mathbf{p}_y)^2/\tau)$.

In this joint optimization, HMA acts as a geometric regularizer to couple the two modalities into a unified manifold, while HPA anchors this coupled pair to the respective identity prototypes, establishing a compact and geometry-consistent embedding space.

### 3.4. Optimization

During training, we optimize HHA with three groups of objectives: (i) baseline identity supervision on ViT features using $\mathcal{L}_{base}$; (ii) identity supervision on HHSA outputs using $\mathcal{L}_{tri}^H$ and $\mathcal{L}_{ce}^H$; (iii) geometry-aware alignment on the Poincaré ball using Hyperbolic Prototype Alignment $\mathcal{L}_{\text{HPA}}$ and Hyperbolic Modality Alignment $\mathcal{L}_{\text{HMA}}$. The overall training objective is:

$$\mathcal{L}_{total} = \mathcal{L}_{base} + \mathcal{L}_{tri}^H + \mathcal{L}_{ce}^H + \lambda_{hma}\mathcal{L}_{\text{HMA}} + \mathcal{L}_{\text{HPA}}, \quad (19)$$

where $\mathcal{L}_{base} = \mathcal{L}_{tri} + \mathcal{L}_{ce}$ denotes the baseline identity losses applied to the ViT features, and $\lambda_{hma}$ controls the strength of the modality alignment term.

## 4. Experiments

### 4.1. Datasets and Evaluation Metrics

We evaluate HHA on two VVI-ReID benchmarks, HITSZ-VCM (Lin et al., 2022) and BUPTCampus (Du et al., 2023). Following common practice, we report CMC and mAP under Infrared-to-Visible (I2V) and Visible-to-Infrared (V2I) protocols.

### 4.2. Implementation Details

We implement HHA with the pretrained CLIP ViT-B/16 visual encoder (Li et al., 2025b; Yu et al., 2026), which is fully fine-tuned end-to-end without frozen layers; the CLIP text encoder is not used during training or inference. Input images are resized to $288 \times 144$ and augmented with random horizontal flipping, padding, cropping, channel erasure, and

*Table 2.* Values of mAP and CMC (%) obtained by our proposed method and the state-of-the-art Re-ID methods on BUPTCampus.

| Methods | Reference | Seq_Len | Infrared-to-Visible | | | | Visible-to-Infrared | | | |
|---|---|---|---|---|---|---|---|---|---|---|
| | | | R@1 | R@5 | R@10 | mAP | R@1 | R@5 | R@10 | mAP |
| MITML(Lin et al., 2022) | CVPR'22 | 6 | 49.1 | 67.9 | 75.4 | 47.5 | 50.2 | 68.3 | 75.7 | 46.3 |
| CLIP-ReID(Li et al., 2023c) | AAAI'23 | 6 | 49.0 | 73.0 | 81.2 | 50.4 | 51.0 | 75.4 | 80.0 | 49.8 |
| TF-CLIP(Yu et al., 2024) | AAAI'24 | 6 | 49.4 | 76.8 | 83.7 | 51.9 | 52.5 | 75.2 | 81.5 | 51.8 |
| VLD(Li et al., 2025b) | TIFS'25 | 6 | 65.3 | 84.9 | **89.7** | 63.5 | 65.8 | 83.0 | 87.9 | 63.0 |
| STHF(Tao et al., 2026) | TCSVT'26 | 6 | 66.7 | 81.8 | 87.0 | 62.1 | 59.8 | 76.8 | 82.8 | 54.6 |
| **HHA(our)** | - | 6 | **67.6** | **85.1** | 88.5 | **64.6** | **68.6** | **85.7** | **90.2** | **64.2** |
| LbA(Park et al., 2021) | ICCV'21 | 10 | 32.1 | 54.9 | 65.1 | 32.9 | 39.1 | 58.7 | 66.5 | 37.1 |
| CAJ(Ye et al., 2021a) | ICCV'21 | 10 | 40.5 | 66.8 | 73.3 | 41.5 | 45.0 | 70.0 | 77.0 | 43.6 |
| AGW(Ye et al., 2021b) | TPAMI'21 | 10 | 36.4 | 60.1 | 67.2 | 37.4 | 43.7 | 64.4 | 73.2 | 41.1 |
| MMN(Zhang et al., 2021) | CVPR'21 | 10 | 40.9 | 67.2 | 74.4 | 41.7 | 43.7 | 65.2 | 73.5 | 42.8 |
| DART(Yang et al., 2022) | CVPR'22 | 10 | 52.4 | 70.5 | 77.8 | 49.1 | 53.3 | 75.2 | 81.7 | 50.5 |
| DEEN(Zhang & Wang, 2023) | CVPR'23 | 10 | 53.7 | 74.8 | 80.7 | 50.4 | 49.8 | 71.6 | 81.0 | 48.6 |
| AuxNet(Du et al., 2023) | TIFS'23 | 10 | 63.6 | 79.9 | 85.3 | 61.1 | 62.7 | 81.5 | 85.7 | 60.2 |
| X-ReID(Yu et al., 2026) | AAAI'26 | 10 | 68.2 | **88.4** | - | **68.5** | 68.8 | 84.8 | - | **65.9** |
| **HHA(our)** | - | 10 | **69.7** | 87.0 | **90.2** | 66.7 | **69.9** | **86.5** | **90.6** | 65.7 |

channel swapping (Ye et al., 2021a). We optimize the model using AdamW with a base learning rate of $2.5 \times 10^{-5}$ on a single NVIDIA H800 GPU. Training is conducted for 60 epochs with a batch size of 32 (2 modalities × 4 identities), where 4 sequences are sampled per identity and 6 frames per sequence. We adopt a 4-epoch warm-up from $1.25 \times 10^{-7}$ to $2.5 \times 10^{-5}$, followed by a step decay of 3 every 10 epochs. The hyperparameters $\lambda_{\mathrm{hma}}$ and $\lambda_p$ are both set to 0.05. Following (Leng et al., 2025; 2024), we fix the hyperbolic curvature $c$ to 1 and use clipping/projection safeguards in log/exp mappings and Möbius operations to keep embeddings within the valid Poincaré ball.

### 4.3. Comparison with State-of-the-Art Methods

In this section, we conducted comparative experiments with other state-of-the-art methods on the HITSZ-VCM and BUPTCampus datasets.

**Evaluation on the HITSZ-VCM Dataset.** Tab. 1 reports results on the HITSZ-VCM dataset under I2V and V2I protocols. With a sequence length of 6, our method achieves 76.0% / 77.5% Rank-1 accuracy and 63.2% / 60.8% mAP, outperforming all existing methods. Compared with recent state-of-the-art approaches such as VLD and X-ReID, our method consistently yields higher Rank-1 and mAP scores under identical or shorter sequence lengths. Increasing the sequence length to 10 further boosts performance to 77.5% / 78.1% Rank-1 accuracy and 64.7% / 62.8% mAP, demonstrating stable performance gains with longer temporal context. These improvements stem from modeling and aligning spatio-temporal features in a unified hyperbolic space.

**Evaluation on the BUPTCampus Dataset.** To evaluate scalability, we conduct experiments on the BUPTCampus benchmark, a larger and more diverse pedestrian dataset.

*Table 3.* Component ablation of HHA on HITSZ-VCM under I2V and V2I protocols, where B denotes the baseline.

| B | HHSA | | GMA | | I2V | | V2I | |
|---|---|---|---|---|---|---|---|---|
| | HGI | DGF | HMA | HPA | R@1 | mAP | R@1 | mAP |
| ✓ | ✗ | ✗ | ✗ | ✗ | 64.4 | 49.8 | 63.6 | 47.7 |
| ✓ | ✓ | ✗ | ✗ | ✗ | 69.4 | 53.2 | 68.4 | 49.6 |
| ✓ | ✓ | ✓ | ✗ | ✗ | 72.0 | 58.3 | 72.7 | 56.9 |
| ✓ | ✓ | ✓ | ✓ | ✗ | 74.1 | 59.5 | 75.9 | 58.1 |
| ✓ | ✓ | ✓ | ✓ | ✓ | 76.0 | 63.2 | 77.5 | 60.8 |

*Table 4.* Ablation on HGI placement in HITSZ-VCM, evaluating insertion frequency and layer position.

| Placement Strategy | I2V | | V2I | |
|---|---|---|---|---|
| | R@1 | mAP | R@1 | mAP |
| *Frequency Ablation* | | | | |
| every layer ($N = 1$) | 73.1 | 58.6 | 74.7 | 57.4 |
| interleaved ($N = 2$) | 76.0 | 63.2 | 77.5 | 60.8 |
| interleaved ($N = 3$) | 73.4 | 59.8 | 76.1 | 59.8 |
| interleaved ($N = 4$) | 73.6 | 59.5 | 76.3 | 58.7 |
| *Position Ablation* | | | | |
| shallow layers ($L_{1\sim6}$) | 72.1 | 59.5 | 72.3 | 56.8 |
| deep layers ($L_{7\sim12}$) | 72.1 | 58.1 | 73.0 | 55.9 |

Tab. 2 reports results under I2V and V2I protocols. With a sequence length of 6, our method achieves 67.6% / 68.6% Rank-1 accuracy and 64.6% / 64.2% mAP, outperforming most existing methods. Compared with recent state-of-the-art approaches such as VLD and X-ReID, our method remains competitive in Rank-1 accuracy and attains comparable or higher mAP under identical sequence length settings. When the sequence length is increased to 10, our method further achieves 69.7% / 69.9% Rank-1 accuracy and 66.7% / 65.7% mAP, indicating consistent gains with longer temporal context. These results validate the effectiveness of

*Table 5.* Geometry Choice of HHSA on HITSZ-VCM.

| HGI Geometry | DGF | | I2V | | V2I | |
|---|---|---|---|---|---|---|
| | E | H | R@1 | mAP | R@1 | mAP |
| Euclidean | ✓ | ✗ | 72.1 | 57.1 | 72.5 | 54.7 |
| Euclidean | ✗ | ✓ | 70.6 | 53.8 | 71.0 | 51.5 |
| Euclidean | ✓ | ✓ | 71.7 | 56.3 | 71.5 | 54.4 |
| Hyperbolic | ✓ | ✗ | 75.6 | 62.5 | 76.6 | 59.6 |
| Hyperbolic | ✗ | ✓ | 74.0 | 60.5 | 75.7 | 58.6 |
| Hyperbolic | ✓ | ✓ | 76.0 | 63.2 | 77.5 | 60.8 |

*Table 6.* Per-frame efficiency comparison. Latency and FPS denote inference time and throughput.

| Method | Params(M) | FLOPs(G) | Latency(ms) | FPS |
|---|---|---|---|---|
| B | 87.54 | 13.96 | 0.85 | 1173.0 |
| HHA | 103.37 | 14.24 | 0.92 | 1089.6 |

hyperbolic modeling on challenging and diverse data.

### 4.4. Ablation Study

We evaluate the contribution of each component in the proposed HHA framework on the HITSZ-VCM dataset. The baseline trains the CLIP vision encoder without hyperbolic modeling and explicit cross-modality alignment. The ablation results on BUPTCampus are reported in Appendix C.

*1) Analysis of HHSA.* We first examine the effect of the proposed HHSA by integrating it into the baseline. As shown in Tab. 3, enabling HHSA (HGI+DGF) brings clear gains under both protocols, improving I2V from 64.4%/49.8% to 72.0%/58.3% (Rank-1/mAP) and V2I from 63.6%/47.7% to 72.7%/56.9%. This validates that HHSA provides a stronger spatio-temporal representation foundation.

**Effect of HGI.** Starting from the baseline, adding HGI alone consistently improves performance (I2V: 64.4%/49.8% to 69.4%/53.2%; V2I: 63.6%/47.7% to 68.4%/49.6%). HGI effectively calibrates frame-wise CLS tokens and alleviates cue entanglement during temporal aggregation. By performing temporal interaction in hyperbolic space, it organizes diverse frame cues more coherently, yielding more stable tracklet representations in both modalities.

**Effect of DGF.** Further incorporating DGF on top of HGI leads to additional improvements (I2V: 69.4%/53.2% to 72.0%/58.3%; V2I: 68.4%/49.6% to 72.7%/56.9%). This shows that DGF is key to distilling identity cues from the hierarchical token memory of HGI. By combining CLS-guided Euclidean and hyperbolic readouts, DGF captures complementary appearance and structural cues, yielding more discriminative embeddings and consistent gains in Rank-1 and mAP.

**Placement Strategy of HGI.** We further investigate the

placement strategy of HGI by evaluating both insertion frequency and layer position, as reported in Tab. 4. For frequency ablation, interleaving HGI every two Transformer layers ($N = 2$) achieves the best performance under both I2V and V2I protocols, outperforming denser ($N = 1$) and sparser ($N \geq 3$) insertions. For position ablation, inserting HGI in interleaved layers consistently outperforms restricting it to either shallow or deep layers. Based on these results, we adopt the interleaved insertion strategy ($N = 2$) as the default configuration.

**Geometry Choice of HHSA.** We further ablate the geometry design of HHSA by switching the attention space in HGI and DGF (Tab. 5). When HGI is implemented in Euclidean space, the performance remains limited regardless of the DGF choice: using Euclidean-only, hyperbolic-only, or dual attentions in DGF yields comparable results (e.g., 72.1%/57.1% vs. 70.6%/53.8% vs. 71.7%/56.3% under I2V), suggesting that Euclidean temporal interaction is insufficient to preserve cue-wise structures. In contrast, replacing HGI with hyperbolic temporal interaction leads to a clear and consistent gain across all DGF settings, improving I2V Rank-1/mAP from 72.1%/57.1% to 75.6%/62.5% even with Euclidean-only DGF, and achieving the best results when DGF fuses both geometries (76.0%/63.2%). A similar trend is observed under V2I (from 72.5%/54.7% to 77.5%/60.8%). These results validate that hyperbolic interaction in HGI is the key factor for effective hierarchical aggregation, while the dual-geometry fusion in DGF further exploits complementary appearance (Euclidean) and structural (hyperbolic) cues for robust video embeddings.

*2) Analysis of GMA.* We further evaluate GMA together with HHSA. As shown in Tab. 3, GMA consistently boosts HHSA from 72.0%/58.3% to 76.0%/63.2% on I2V and from 72.7%/56.9% to 77.5%/60.8% on V2I, validating that GMA complements HHSA.

**Effect of HMA.** We first enable HMA to alleviate the visible-infrared modality gap on the Poincaré ball by pulling modality centroids closer and reducing identity-conditioned offsets. With HMA, HHSA improves from 72.0%/58.3% to 74.1%/59.5% under the I2V protocol, and from 72.7%/56.9% to 75.9%/58.1% under the V2I protocol.

**Effect of HPA.** After establishing modality-consistent representations, we activate HPA to pull embeddings toward shared identity prototypes, enhancing discrimination under the same hyperbolic metric. With HPA, the model achieves the best results in Tab. 3 (76.0%/63.2% on I2V and 77.5%/60.8% on V2I). This suggests that HMA and HPA are complementary: HMA reduces the visible-infrared shift, while HPA further enforces prototype-centered discrimination for cross-modality retrieval.

*3) Model Complexity.* We further report model complexity

*Table 7.* Tree-likeness analysis using normalized $\delta$-hyperbolicity. Smaller is better.

| Metric | Euclidean | HHA | Change |
|---|---|---|---|
| Mean $\delta_{\max}$ | 0.0627 | 0.0419 | -33.2% |
| Median $\delta_{\max}$ | 0.0623 | 0.0400 | -35.8% |
| Mean $\delta_{\mean}$ | 0.01021 | 0.00456 | -55.3% |

*Table 8.* Curvature sensitivity analysis on HITSZ-VCM.

| $c$ | I2V | | V2I | |
|---|---|---|---|---|
| | R@1 | mAP | R@1 | mAP |
| 0.5 | 72.2 | 59.0 | 72.4 | 55.9 |
| 1.0 | 76.0 | 63.2 | 77.5 | 60.8 |
| 1.5 | 75.5 | 61.3 | 77.6 | 59.4 |
| 2.0 | 74.7 | 59.5 | 76.0 | 57.4 |

*Table 9.* Comparison with DS-VReID adapted to VVI-ReID.

| Method | I2V | | V2I | |
|---|---|---|---|---|
| | R@1 | mAP | R@1 | mAP |
| DS-VReID | 70.4 | 57.1 | 71.6 | 55.1 |
| HHA | 76.0 | 63.2 | 77.5 | 60.8 |

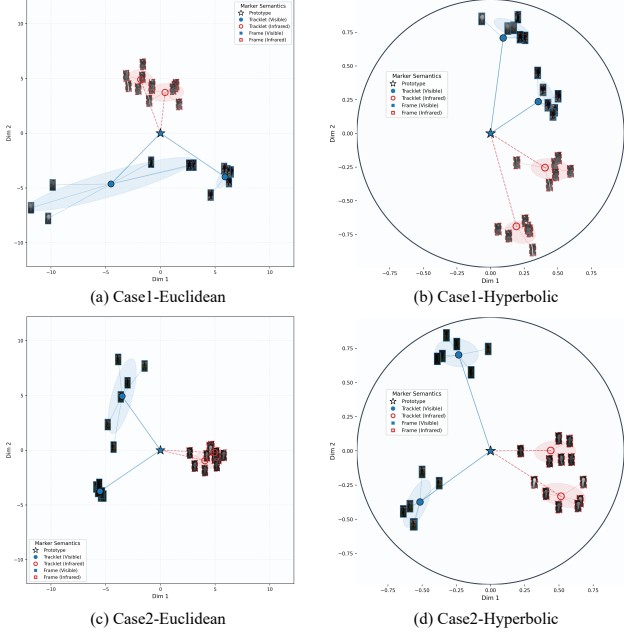

(a) Case1-Euclidean  (b) Case1-Hyperbolic
(c) Case2-Euclidean  (d) Case2-Hyperbolic

*Figure 3.* Identity-centered visualization of Euclidean counterpart and HHA. HHA shows more compact prototype-centered organization with less cue crowding.

in Tab. 6. Compared with the baseline, HHA introduces a modest increase in parameters and per-frame computation, while maintaining real-time throughput.

### 4.5. Further Analysis of Hyperbolic Modeling

We further analyze HHA on HITSZ-VCM in terms of tree-likeness, curvature sensitivity, identity-centered cue organization, and comparison with DS-VReID.

**Tree-likeness of Learned Representations.** HHA aims to learn a less crowded and more tree-like representation space for organizing diverse temporal variations. We adopt normalized $\delta$-hyperbolicity to analyze learned identity-level point sets, where smaller values indicate a more tree-like or hyperbolic structure (Chami et al., 2019). As shown in Tab. 7, HHA substantially reduces all $\delta$ values compared with its Euclidean counterpart, supporting the geometric motivation of hyperbolic temporal aggregation.

**Curvature Sensitivity.** We further study the influence of the curvature parameter $c$ in the Poincaré ball. As shown in Tab. 8, HHA remains reasonably stable within a practical range of $c$ on HITSZ-VCM. The setting $c = 1.0$ achieves the best overall performance and is therefore used as the default setting in all experiments.

**Comparison with DS-VReID.** We compare HHA with DS-VReID (Leng et al., 2025), a recent dual-space method for visible-only video ReID. After adapting DS-VReID to VVI-ReID, HHA consistently performs better on HITSZ-VCM, as shown in Tab. 9. This indicates that HHA is better suited

to VVI-ReID, as it performs hyperbolic temporal aggregation directly on frame-level CLS tokens while further enforcing geometry-aware visible-infrared alignment.

**Visualization of Hyperbolic Cue Organization.** Fig. 3 visualizes how frame-level cues are organized around identity prototypes in the Euclidean counterpart and HHA. Compared with the Euclidean counterpart, where visible/infrared tracklets are more scattered and frame-level cues are crowded, HHA forms more compact prototype-centered structures. This indicates that hyperbolic modeling better organizes spatio-temporal cues under pose, viewpoint, and modality variations around shared identity prototypes.

## 5. Conclusion

In this paper, we propose Hyperbolic Hierarchical Alignment (HHA), a geometry-aware framework for VVI-ReID that performs hierarchical spatio-temporal modeling and cross-modality alignment on the Poincaré ball. HHA leverages HHSA, where HGI captures hierarchical temporal interactions and DGF fuses Euclidean and hyperbolic attentions to aggregate diverse frame cues into stable identity representations, and further introduces GMA that combines HMA and HPA to reduce visible-infrared discrepancy and strengthen prototype-centered discrimination. Experiments on HITSZ-VCM and BUPTCampus demonstrate the effectiveness and competitiveness of HHA.

## Acknowledgements

This work was supported in part by the National Natural Science Foundation of China under Grants No. 62221005, 62472060, U22A2096 and U23A20318, in part by the Science and Technology Innovation Key R&D Program of Chongqing under Grant No. CSTB2023TIAD-STX0016, in part by the Natural Science Foundation of Chongqing under Grants No. CSTB2024NSCQ-QCXMX0060, in part by Chongqing University of Posts and Telecommunications Ph.D. Innovative Talents Project under Grant No. BYJS202401, and in part by the Chongqing Postgraduate Research and Innovation Project under Grants No. CYS240416.

## Impact Statement

The proposed VVI-ReID framework can support public safety applications in low-light scenarios by matching visible and infrared video tracklets across cameras, such as missing-person search, trajectory analysis, and event investigation.

Meanwhile, person re-identification may raise privacy and surveillance concerns if misused. Responsible deployment requires lawful usage, data protection, transparent governance, and appropriate oversight.

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

# A. Hyperbolic Geometry Preliminaries

Hyperbolic space is a non-Euclidean Riemannian manifold with constant negative curvature. It admits several equivalent representations, including the Lorentz (hyperboloid) model, the Poincaré ball and half-plane models, and the Klein model. In this work, we adopt the Poincaré ball model for its conformal relation to the Euclidean metric and its computational convenience.

**Poincaré ball model.** The Poincaré ball model $\mathbb{B}_c^n$ is a manifold representing an $n$-dimensional ball equipped with a Riemannian metric $\mathfrak{g}_x^B$. It is conformal to the Euclidean metric $\mathfrak{g}_x^E$ with conformal factor $\lambda_x^c = \frac{2}{1-c|x|^2}$, where $c > 0$ and the sectional curvature is $-c$. Accordingly, the metric can be written as $\mathfrak{g}_x^B = (\lambda_x^c)^2 \mathfrak{g}_x^E$. Formally, the manifold is defined as:

$$\mathbb{B}_c^n = \{x \in \mathbb{R}^n : c\|x\|^2 < 1\} \quad \text{(i.e., } \|x\| < 1/\sqrt{c}), \tag{20}$$

where $\|\cdot\|$ denotes the Euclidean norm. For any $x, y \in \mathbb{B}_c^n$, the hyperbolic distance between $x, y$ is given by:

$$d_c(x, y) = \frac{1}{\sqrt{c}} \text{arcosh}\Big(1 + \frac{2c\|x - y\|^2}{(1 - c\|x\|^2)(1 - c\|y\|^2)}\Big). \tag{21}$$

**Tangent space.** Since $\mathbb{B}_c^n$ is an open subset of $\mathbb{R}^n$, the tangent space at any point $x$ can be naturally identified with $\mathbb{R}^n$. The metric $\mathfrak{g}_x^B$ induces the following inner product at $x$:

$$\langle u, v \rangle_x = (\lambda_x^c)^2 \langle u, v \rangle_E, \quad u, v \in \mathcal{T}_x \mathbb{B}_c^n. \tag{22}$$

In particular, the induced norm satisfies $\|v\|_x = \lambda_x^c \|v\|$, and thus $\mathcal{T}_x \mathbb{B}_c^n \cong \mathbb{R}^n$.

**Exponential and logarithmic maps.** Mappings between the manifold and its tangent space are given by the exponential and logarithmic maps. The exponential map $\exp_x^c : \mathcal{T}_x \mathbb{B}_c^n \to \mathbb{B}_c^n$ is defined as:

$$\exp_x^c(v) = x \oplus_c \frac{1}{\sqrt{c}} \tanh\Big(\frac{\sqrt{c}\lambda_x^c \|v\|}{2}\Big) \frac{v}{\|v\|}. \tag{23}$$

Conversely, the logarithmic map $\log_x^c : \mathbb{B}_c^n \to \mathcal{T}_x \mathbb{B}_c^n$, is given by:

$$\log_x^c(y) = \frac{2}{\sqrt{c}\lambda_x^c} \tanh^{-1}(\sqrt{c}\|y \ominus_c x\|) \frac{y \ominus_c x}{\|y \ominus_c x\|}, \tag{24}$$

where $x, y \in \mathbb{B}_c^n$ and $v \in \mathcal{T}_x \mathbb{B}_c^n$. Here, $\oplus_c : \mathbb{B}_c^n \times \mathbb{B}_c^n \to \mathbb{B}_c^n$ denotes the Möbius addition operator:

$$x \oplus_c y = \frac{(1 + 2c\langle x, y \rangle + c\|y\|^2)x + (1 - c\|x\|^2)y}{1 + 2c\langle x, y \rangle + c^2\|x\|^2\|y\|^2}, \tag{25}$$

and the corresponding subtraction is defined as:

$$x \ominus_c y = x \oplus_c (-y). \tag{26}$$

Accordingly, the Möbius summation can be written as:

$$\bigoplus_{i=1}^N x_i = (\cdots((x_1 \oplus_c x_2) \oplus_c x_3) \cdots) \oplus_c x_N. \tag{27}$$

**Möbius matrix-vector and scalar multiplication.** To implement transformations on the Poincaré ball, the tangential method is commonly adopted. Specifically, Möbius matrix-vector multiplication is defined as:

$$\mathbf{M} \otimes_c x = \frac{1}{\sqrt{c}} \tanh\Big(\frac{\|\mathbf{M}x\|}{\|x\|} \text{artanh}(\sqrt{c}\|x\|)\Big) \frac{\mathbf{M}x}{\|\mathbf{M}x\|} = \exp_{\mathbf{o}}^c\big(\mathbf{M} \log_{\mathbf{o}}^c(x)\big), \tag{28}$$

where $x \in \mathbb{B}_c^n$, $\mathbf{M} \in \mathbb{R}^{d \times n}$, and $\mathbf{o}$ denotes the origin in the Poincaré ball $\mathbb{B}_c^n$.

Similarly, for a scalar $\alpha \in \mathbb{R}$, scalar Möbius multiplication is defined as:

$$\alpha \otimes_c x = \frac{1}{\sqrt{c}} \tanh\big(\alpha \, \text{artanh}(\sqrt{c}\|x\|)\big) \frac{x}{\|x\|} = \exp_{\mathbf{o}}^c\big(\alpha \log_{\mathbf{o}}^c(x)\big). \tag{29}$$

## B. Theoretical Motivation: Why Hyperbolic Space?

To provide a rigorous foundation for the proposed Hyperbolic Hierarchical Alignment (HHA), we analyze the geometric limitations of Euclidean embeddings in representing complex video dynamics and modality gaps.

A video tracklet $\mathcal{V}$ can be structurally modeled as a hierarchical organization: the global identity serves as the root, while diverse temporal variations (e.g., pose changes, occlusions, and motion blur) form branching realizations. Let $b$ be the branching factor and $d$ be the depth of this hierarchical temporal tree. The number of leaf nodes, representing the complexity of visual cues, grows as $O(b^d)$.

In an $n$-dimensional Euclidean space $\mathbb{R}^n$, the volume of a ball with radius $r$ scales polynomially:

$$V_E(r) = \frac{\pi^{n/2}}{\Gamma(\frac{n}{2}+1)} r^n \propto r^n. \tag{30}$$

Conversely, in the Poincaré ball $\mathbb{B}_c^n$ with constant negative curvature $-c$, the volume grows exponentially with the radius $r$:

$$V_H(r) = \frac{(n-1)\pi^{n/2}}{\Gamma(\frac{n}{2}+1)} \int_0^r \sinh^{n-1}(\sqrt{c}s)ds \propto e^{(n-1)\sqrt{c}r}. \tag{31}$$

This exponential capacity allows $\mathbb{B}_c^n$ to embed tree-like video structures with lower metric distortion $\varepsilon$. In contrast, embedding such structures into $\mathbb{R}^n$ requires the distortion to grow exponentially with the depth $d$. This geometric mismatch in Euclidean space forces heterogeneous frame features into a crowded embedding space, leading to the cue crowding phenomenon where discriminative identity details are prone to being collapsed.

## C. Additional Ablation Studies

*Table 10.* Ablation study of HHA components on BUPTCampus. Starting from the baseline (B), we progressively add HGI, DGF, HMA, and HPA. Results are reported under the I2V and V2I protocols. "R@1" denotes Rank-1.

| B | HHSA | | GMA | | *Infrared-to-Visible* | | *Visible-to-Infrared* | |
|---|------|------|------|------|------|------|------|------|
| | HGI | DGF | HMA | HPA | R@1 | mAP | R@1 | mAP |
| ✓ | ✗ | ✗ | ✗ | ✗ | 40.8 | 42.5 | 40.8 | 43.6 |
| ✓ | ✓ | ✗ | ✗ | ✗ | 58.2 | 57.7 | 58.6 | 56.1 |
| ✓ | ✓ | ✓ | ✗ | ✗ | 63.0 | 62.0 | 64.4 | 62.6 |
| ✓ | ✓ | ✓ | ✓ | ✗ | 67.1 | 64.3 | 65.0 | 61.0 |
| ✓ | ✓ | ✓ | ✓ | ✓ | 67.6 | 64.6 | 68.6 | 64.2 |

Tab. 10 presents an ablation study on BUPTCampus, summarizing the contribution of each component. Starting from the baseline, enabling HHSA consistently improves retrieval under both infrared-to-visible and visible-to-infrared protocols. In particular, adding HGI brings a gain of 17.4% / 15.2% in Rank-1 / mAP in the infrared-to-visible protocol and 17.8% / 12.5% in the visible-to-infrared protocol, and further incorporating DGF yields an additional 4.8% / 4.3% in the infrared-to-visible protocol and 5.8% / 6.5% in the visible-to-infrared protocol, validating the effectiveness of hyperbolic hierarchical aggregation. Building upon HHSA, introducing HMA further boosts performance by reducing the modality gap on the manifold, improving Rank-1 by 4.1% in the infrared-to-visible protocol and 0.6% in the visible-to-infrared protocol. Finally, adding HPA achieves the best results by pulling features toward their identity prototypes and thereby enhancing discrimination, with an overall improvement of 26.8% / 22.1% in the infrared-to-visible protocol and 27.8% / 20.6% in the visible-to-infrared protocol over the baseline, demonstrating the effectiveness of hyperbolic spatio-temporal modeling together with geometry-aware visible-infrared alignment.

## D. Hyper-Parameter Analysis and Qualitative Visualization

### D.1. Parameter Analysis

We analyze the sensitivity of two hyper-parameters, $\lambda_p$ and $\lambda_{hma}$, on HITSZ-VCM under the visible-to-infrared protocol. As shown in Fig. 4(a-b), we vary one parameter while fixing the other to $0.05$ and report Rank-1 and mAP. Both parameters

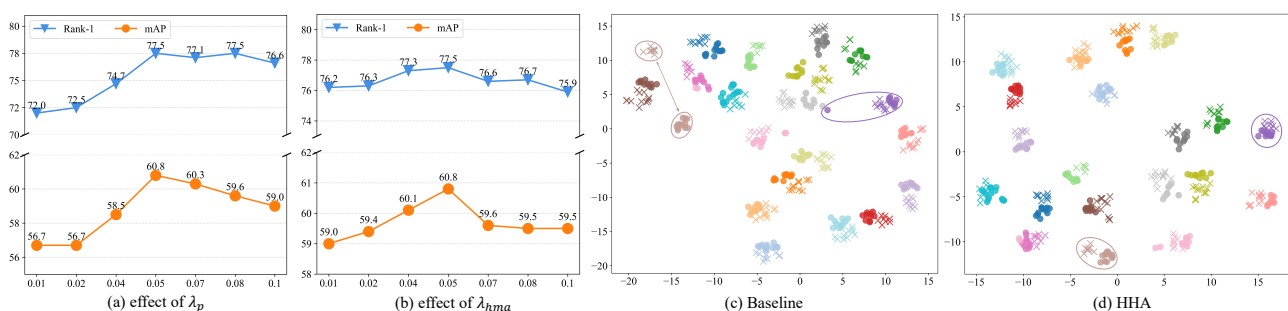

*Figure 4.* Hyper-parameter analysis and embedding visualization on HITSZ-VCM. (a-b) Performance under different $\lambda_p$ and $\lambda_{hma}$ in the visible-to-infrared setting; setting both to $0.05$ yields the best Rank-1/mAP. (c-d) t-SNE visualization of cross-modality embeddings for the baseline and HHA. Circles and crosses denote visible and infrared modalities, respectively, and colors indicate identities.

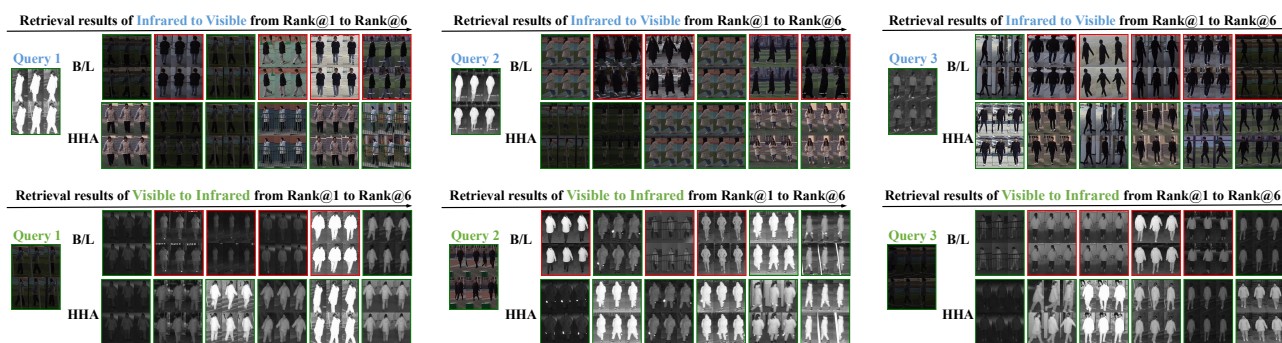

*Figure 5.* Qualitative cross-modality retrieval results on HITSZ-VCM. Top-6 matches (Rank@1-Rank@6) are shown for Infrared-to-Visible (top) and Visible-to-Infrared (bottom) queries. Green and red boxes indicate correct and incorrect matches, respectively.

exhibit a consistent unimodal trend: performance improves when the corresponding constraint becomes effective, and degrades when it is over-weighted due to over-regularization.

Specifically, increasing $\lambda_p$ from $0.01$ to $0.05$ yields clear gains on both Rank-1 and mAP, while further increasing it leads to a noticeable drop in Fig. 4(a). A similar trend is observed for $\lambda_{hma}$ in Fig. 4(b), where setting it to $0.05$ achieves the highest Rank-1 and mAP, and larger values gradually hurt performance. Therefore, we set $\lambda_p = 0.05$ and $\lambda_{hma} = 0.05$ in all experiments.

### D.2. Qualitative Visualization

We provide qualitative analysis via t-SNE visualization and cross-modality retrieval examples under infrared-to-visible and visible-to-infrared settings to illustrate representation quality and retrieval performance.

**Feature Embedding Analysis.** Fig. 4 (c-d) visualizes cross-modality embeddings on HITSZ-VCM, where circles and crosses denote visible and infrared samples, and colors indicate identities. In Fig. 4(c), the baseline exhibits clear modality-induced fragmentation for the same identity. For instance, the light-brown identity marked on the left is split into two distant sub-clusters, with visible and infrared samples forming separated groups, suggesting that modality bias dominates the neighborhood structure and weakens identity compactness. A similar issue is observed for the purple identity on the right, where a separated branch and an outlying point appear, indicating unstable tracklet representations. In contrast, Fig. 4(d) shows that HHA produces tighter and more coherent identity clusters. The previously separated light-brown samples become merged into a single compact cluster with improved cross-modality consistency, and the purple outlier is absorbed back toward its identity center, reducing unnecessary dispersion. These observations imply that HHSA, via HGI and DGF, aggregates diverse frame-wise cues into more stable tracklet embeddings, while GMA further suppresses modality-dependent offsets, leading to more geometry-consistent cross-modality clustering.

**Retrieval Results Analysis.** Fig. 5 illustrates qualitative cross-modality retrieval results under infrared-to-visible and visible-to-infrared settings for representative queries. The baseline (B/L) frequently ranks incorrect identities among the top

results (Rank@1-6), reflecting that modality discrepancy and background or appearance bias dominate retrieval. With HHA, correct matches (green boxes) appear more consistently at higher ranks in both directions, and hard negatives caused by pose changes, occlusion, and background clutter are notably reduced. These examples confirm that HHA benefits from hyperbolic hierarchical modeling, which organizes diverse frame cues with lower distortion and enforces geometry-consistent visible-infrared alignment on the Poincaré ball, leading to more stable modality-invariant identity representations and more reliable cross-modality retrieval.

