# OpenReview forum: "Hyperbolic Hierarchical Alignment for Video-Based Visible-Infrared Person Re-Identification"
_ICML.cc/2026/Conference — ICML 2026 regular_

### Official Review · Reviewer_DwM4 · 2026-03-06

**Soundness:** 3
**Presentation:** 3
**Significance:** 2
**Originality:** 4
**Overall Recommendation:** 4
**Confidence:** 4

**Summary:**

This paper addresses video-based visible-infrared person re-identification  by introducing Hyperbolic Hierarchical Alignment (HHA), a framework that models video tracklets in hyperbolic space. The key insight is that temporal variations within a person’s video sequence—such as those caused by occlusion or pose changes—naturally form a hierarchical, tree-like structure, which is better preserved in hyperbolic geometry due to its exponential growth property. To this end, the authors design a unified pipeline on the Poincaré ball: a Hyperbolic Hierarchical Spatio-Temporal Aggregator (HHSA) captures intra-sequence dynamics by fusing Euclidean and hyperbolic attention, while a Geometry-Aware Modality Alignment (GMA) module aligns visible and infrared features through modality centroids and shared identity prototypes in hyperbolic space. The method achieves state-of-the-art results on standard VVI-ReID benchmarks, demonstrating the effectiveness of geometric-aware modeling for cross-modal video matching.

**Compliance With Llm Reviewing Policy:**

Affirmed.

**Final Justification:**

This paper proposes a original framework for VVI-ReID. My initial concerns regarding optimization stability, computational overhead, and single-modal generalizability were addressed in the rebuttal with new experiments and clear technical clarifications. The authors' responses strengthened the overall soundness and practical significance of the work. As my concerns have been resolved, I maintain my initial positive assessment.

**Key Questions For Authors:**

1.Could you provide a comparison between hyperbolic and spherical embeddings for this task? While hyperbolic space is argued to be optimal for hierarchical data, spherical geometry also offers non-Euclidean structure and has been used in representation learning. Was a spherical variant of HHA explored during development? If not, what theoretical or empirical reasons led to the exclusive choice of hyperbolic geometry?
2. What is the computational overhead of hyperbolic operations during inference, and how does it compare to a Euclidean baseline?

**Limitations:**

The authors do not explicitly discuss the limitations of their work or its potential negative societal impact. While the paper presents a technically strong and innovative approach, it lacks a section addressing key constraints such as computational overhead, sensitivity to hyperparameters, or failure modes under real-world conditions. Additionally, no discussion is provided on how the method might be misused in surveillance systems—particularly given its application to cross-modal person re-identification, which raises privacy concerns.

**Strengths And Weaknesses:**

Strengths：
1.The paper is the first to apply hyperbolic geometry to VVI-ReID, offering a principled way to model the hierarchical temporal structure of tracklets that Euclidean spaces struggle to preserve.
2. Experiments on HITSZ-VCM and BUPTCampus show consistent state-of-the-art performance. Ablation studies systematically verify each module’s contribution.
3. The problem setup, geometric intuition, and technical design are communicated clearly with helpful illustrations.

Weaknesses:
1. The use of identity prototypes in the Poincaré ball is effective empirically, but the paper does not address how prototype initialization, update rules, or convergence behave under hyperbolic metric constraints. In Euclidean space, prototype-based losses are well-understood; their hyperbolic counterparts may suffer from gradient vanishing near the boundary or instability due to curvature.
2. Training complexity and optimization challenges glossed over: Optimizing deep networks in hyperbolic space requires careful handling of numerical stability. The paper provides minimal details on training tricks, convergence behavior, or sensitivity to curvature hyperparameter  c , making reproduction and extension difficult.
3. While promising, the approach is evaluated only on two VVI-ReID datasets. The HHA method does not appear to be specifically designed for cross-modal video learning; therefore, it can be evaluated on single-modal video datasets to enhance the generalizability of the proposed approach in this work.

---

> ### Author Rebuttal · Authors · 2026-03-31
>
> Thank you for your constructive review. Below, we respond point by point (W: Weakness, Q: Question, L: Limitation).
>
> **[W1,W2]** Thank you for raising this important point. We agree that prototype learning and optimization in hyperbolic space require careful treatment, and the current manuscript did not describe these details clearly enough.
>
> In our implementation, identity prototypes are initialized with small norm in the tangent space at the origin and optimized in this Euclidean tangent space, rather than being directly updated as free points on the Poincaré ball. During forward computation, they are mapped to the manifold via the exponential map and projected back to the valid region, which helps mitigate boundary-related instability. More generally, we keep the computation geometry-consistent and apply explicit clipping/projection safeguards so that features remain strictly inside the valid ball region. Hyperbolic operations are introduced only in the temporal interaction and structural aggregation modules, rather than throughout the entire backbone. For reproducibility, we provide pseudo-code (see our response to **Reviewer 8f8T, Q2**), and core implementation in image form via **https://anonymous.4open.science/r/anon-figures-F024**. The code will be publicly available.
>
> We additionally include the curvature sensitivity analysis below. The results show that HHA remains reasonably stable across a practical range of c, with c=1 providing the best overall trade-off between performance and optimization stability in our experiments. This is also consistent with prior hyperbolic representation learning work, where c=1 is a common and stable default choice. We will clarify the prototype initialization, update rule, and numerical stabilization details in the revision.
>
> Curvature sensitivity results.
> c|VCM I2V (R1/mAP)|VCM V2I (R1/mAP)|BUPT I2V (R1/mAP)|BUPT V2I (R1/mAP)
> -|-|-|-|-
> 0.5|72.2/59.0|72.4/55.9|60.7/57.7|59.2/54.3
> 1.0|76.0/63.2|77.5/60.8|67.6/64.6|68.6/64.2
> 1.5|75.5/61.3|77.6/59.4|66.5/63.9|69.0/65.4
> 2.0|74.7/59.5|76.0/57.4|66.3/63.0|67.4/64.0
>
> **[W3]** Thank you for the helpful suggestion. We further evaluated the modality-agnostic HHSA (HGI + DGF) on the single-modal video ReID benchmarks MARS and LS-VID, while removing the cross-modal-specific GMA. As shown below, HHSA consistently improves the baseline and slightly outperforms DS-VReID, a recent hyperbolic video ReID method, on both datasets, indicating that the benefit of HHSA is not limited to the VVI-ReID setting.
>
> Single-modal transfer results without GMA.
> Method|MARS (R1/mAP)|LS-VID (R1/mAP)
> -|-|-|
> DS-VReID[1]|92.3/87.6|88.2/78.7
> B + HHSA|92.4/88.8|89.9/82.6
>
> [1] Leng et al. Dual-space video person re-identification. IJCV, 2025.
>
> **[Q1]** Thank you for this question. Our choice of hyperbolic space is not simply because it is non-Euclidean, but because the structure we model is hierarchy-aware spatio-temporal organization, where diverse frame-level variations are progressively aggregated into sequence-level representations. Due to its negative curvature, hyperbolic space exhibits exponential volume growth, which provides more capacity to accommodate this progressively expanding structure with lower distortion. By contrast, spherical geometry has positive curvature and a closed structure that is less naturally matched to this setting.
>
> We did not systematically explore a spherical variant in this work. While this would be an interesting comparison, it is not a lightweight substitution and would require redesign of manifold mappings, distance/similarity computation, transformations, attention aggregation, and prototype alignment. We will clarify this motivation in the revision and treat spherical variants as future work.
>
> **[Q2]** We compare full HHA with two references: the Euclidean baseline, i.e., the original backbone without the added modules, and a Euclideanized counterpart, which keeps the same overall architecture but replaces the hyperbolic layers with standard Transformer layers and removes the hyperbolic branch in DGF. The inference comparison is shown below. The results indicate that the added inference overhead of HHA over the Euclideanized counterpart is modest.
>
> Method|Params (M)|FLOPs/frame (G)|Latency/frame (ms)|FPS|Inference Memory (MiB)|
> -|-|-|-|-|-
> Euclidean baseline|87.54|13.96|0.85|1173.0|17378
> Euclideanized counterpart|101.01|14.23|0.88|1140.4|24186
> Full HHA|103.37|14.24|0.92|1089.6|25172
>
> **[L1]** We agree that the current limitations and impact statement is too brief. In the revision, we will make it more explicit by briefly discussing the main technical limitations and the potential privacy/misuse concerns of VVI-ReID.

---

> > ### Author Rebuttal · Reviewer_DwM4 · 2026-04-02
> >
> > I appreciate the authors' clarifications, which have resolved the majority of my initial concerns. Accordingly, I will keep my initial positive score.

---

> > > ### Author Response · Authors · 2026-04-02
> > >
> > > Thank you for your thoughtful review and for considering our rebuttal. We are glad that our clarifications addressed your concerns, and we appreciate your positive assessment.

---

### Official Review · Reviewer_GWrv · 2026-03-09

**Soundness:** 3
**Presentation:** 2
**Significance:** 3
**Originality:** 2
**Overall Recommendation:** 4
**Confidence:** 4

**Summary:**

This paper studies video-based visible-infrared person re-identification (VVI-ReID), where the main challenge is to learn identity-discriminative video representations despite both temporal variation within a tracklet and the modality gap between visible and infrared data. The paper argues that existing methods largely rely on Euclidean geometry, which is not well suited to representing the hierarchical, branching structure induced by temporal variations such as occlusion, pose change, viewpoint change, and motion blur. To address this, the authors propose Hyperbolic Hierarchical Alignment (HHA), an end-to-end framework that performs both spatio-temporal modeling and cross-modality alignment on the Poincaré ball. The method consists of two main parts: HHSA, which includes Hyperbolic Geometry Interaction (HGI) and Dual-Geometry Fusion (DGF) for hierarchical temporal aggregation, and GMA, which includes Hyperbolic Modality Alignment (HMA) and Hyperbolic Prototype Alignment (HPA) for geometry-aware cross-modality alignment and prototype-centered discrimination. Experiments on HITSZ-VCM and BUPTCampus report state-of-the-art performance, supported by ablations on the contribution of each component and on the geometry choices used in the model.

**Compliance With Llm Reviewing Policy:**

Affirmed.

**Final Justification:**

The framework is clearly presented, and the promised revisions will improve the manuscript.
The paper presents a sound geometric perspective for VVI-ReID, supported by thorough experiments. While the scope focuses solely on visible-infrared modalities, the underlying idea — that hyperbolic geometry better preserves hierarchical temporal structure for cross-modal video matching — is well motivated. In line with current trends, examining this method on other modalities would be encouraging; nevertheless, I consider this a solid work.

**Key Questions For Authors:**

1. How sensitive is the method to the curvature parameter 𝑐? The current implementation fixes 𝑐=1.
2. How does HHA compare directly with DS-VReID (Leng et al., 2025)? DS-VReID is the closest methodological predecessor, using dual Euclidean-hyperbolic space for video ReID. The architectural similarity (dual-space aggregation, hyperbolic temporal modeling) suggests HHA may be an extension. I would kindly ask the authors to compare against DS-VReID adapted for cross-modality.
3. How does the authors' HGI mechanism explicitly enforce or model a hierarchy between the global identity and temporal variations? The attention mechanism computes similarity between frame pairs, which seems to capture a "flat" structure. Can the authors provide a more detailed explanation or a toy example illustrating how the hyperbolic distances and learned attention weights organize frames into a tree-like structure with the identity as the root?
4. The paper defines ⊗𝑐 in Appendix A as Möbius matrix-vector multiplication, yet Eq. (4) and Eq. (14) use
⊗𝑐 with 𝛼 being a scalar attention weight. It is therefore unclear whether the intended operation is scalar Möbius multiplication followed by iterative Möbius addition, a tangent-space weighted average, or an approximation to a Fréchet/gyro-barycenter.

**Limitations:**

yes.

**Strengths And Weaknesses:**

Strengths:
- The paper identifies a genuine and interesting limitation in existing VVI-ReID methods: the failure to account for the hierarchical nature of video data (identity vs. variations). Framing this as a geometric mismatch with Euclidean space is a compelling motivation.
- The proposed HHA consistently outperforms a wide range of state-of-the-art methods on two challenging benchmarks. The improvements are significant and clearly demonstrated in Tables 1 and 2.

Weaknesses:
- The introduction explicitly claims that a tracklet forms a branching hierarchy around an identity center, that Euclidean aggregation causes “structural collapse,” and that HGI preserves “branching structures” and “identity-variation structure.” However, the actual HGI module is a hyperbolic attention block over frame-level CLS tokens, where attention weights are computed from pairwise hyperbolic distances and then aggregated with Möbius operations. The method supports a claim like hyperbolic temporal aggregation with multi-layer memory, but it does not directly realize the stronger claim of explicitly modeling an "identity-centered hierarchy".
- A large part of the paper’s argument is that Euclidean geometry distorts hierarchical cue structure, while hyperbolic geometry preserves it with lower distortion. However, the evidence provided is mostly indirect: benchmark gains, module ablations, and a conceptual figure. There is a lack of hierarchy-preservation analysis, manifold visualization showing reduced cue crowding, and no quantitative evidence that the learned embeddings actually exhibit the claimed branching structure.
- The method seems to share a similar intuition with DS-VReID. Leng et al. (2025) already introduced a dual-space formulation for video person ReID, using Euclidean and hyperbolic spaces for different representation purposes. Although HHA adapts this idea to the cross-modality setting, it would be helpful for the authors to more clearly articulate the novelty beyond this extension.
- The curvature setting would benefit from further discussion. The paper fixes c=1. Given that hyperbolic representations can be sensitive to curvature choice, additional analysis on this design decision would make the method more convincing.

---

> ### Author Rebuttal · Authors · 2026-03-31
>
> Thank you for your thoughtful comments. We respond point by point below (W: Weakness, Q: Question).
>
> **[W1,Q3]** Thank you for this comment. We agree that our original hierarchy claim was too strong. HGI does not explicitly construct an identity-rooted tree, so attributing an explicit identity-centered hierarchy to HGI alone is not accurate. Our intended claim is weaker: HHSA provides a hyperbolic inductive bias for better organizing frame-level temporal variations into a sequence representation, while GMA further aligns these representations across modalities and toward shared identity prototypes.
>
> **HGI clarification:** After frame-level CLS tokens are mapped into hyperbolic space, HGI computes attention weights from pairwise hyperbolic distances. Owing to the negative-curvature geometry, this provides a stronger inductive bias for organizing diverse temporal cues with lower crowding than Euclidean aggregation. Multi-layer outputs are preserved as hierarchical memory, allowing these cues to continue influencing the final sequence representation rather than being immediately compressed. Thus, HGI mainly provides a more structured organization of frame-level temporal variations, rather than explicitly constructing an identity-rooted tree.
>
> **[W2]** Thank you for this important comment. We agree that the hierarchy claim needs more direct evidence, so we compare HHA with its Euclidean counterpart (replacing HGI with Euclidean attention) through both quantitative analysis and identity-level visualization.
>
> **Quantitative analysis**: We use $\delta$-hyperbolicity to measure the tree-likeness of the representation space, where smaller values indicate a more tree-like structure. For each identity, we form a local point set spanning both modalities and multiple tracklets/frames, and compute normalized $\delta$-hyperbolicity on this set.
>
> Metric|Euclidean|HHA|Relative change
> -|-|-|-
> Mean normalized $\delta_{\max}$|0.0627|0.0419|-33.2%
> Median normalized $\delta_{\max}$|0.0623|0.0400|-35.8%
> Mean normalized $\delta_{\mathrm{mean}}$|0.01021|0.00456|-55.3%
>
> **Visualization analysis:** We also provide identity-level visualizations at **https://anonymous.4open.science/r/anon-figures-F024**. Compared with its Euclidean counterpart, HHA shows clearer prototype-centered organization and reduced cue crowding.
>
> This does not prove that HHA learns an explicit tree, but it provides more direct evidence that its learned space is more tree-like and less crowded than the Euclidean counterpart.
>
> **[W3,Q2]** Thank you for this comment. DS-VReID is designed for visible-only VReID, and its dual-space modeling is built on DPGC, HDA, and DSF over explicitly constructed graphs. In contrast, HHA is designed for VVI-ReID: it performs hyperbolic temporal aggregation directly on frame-level CLS tokens through HGI + DGF, and further conducts geometry-aware visible-infrared alignment through GMA. Thus, HHA is not simply a transfer of DS-VReID, but a framework designed for VVI-ReID.
>
> Following your suggestion, we adapted DS-VReID to VCM and BUPTCampus for VVI-ReID, modifying the data pipeline and training objective while preserving its original dual-space design for a fair comparison. Even so, it still underperforms HHA, as shown below:
> Method|VCM I2V (R1/mAP)|VCM V2I (R1/mAP)|BUPT I2V (R1/mAP)|BUPT V2I (R1/mAP)
> -|-|-|-|-
> DS-VReID|70.4/57.1|71.6/55.1|59.8/59.2|58.0/57.3
> HHA|76.0/63.2|77.5/60.8|67.6/64.6|68.6/64.2
>
> **[W4,Q1]** Thank you for this question. We evaluated multiple curvature values and found that HHA remains stable across a practical range of c, with c=1 giving the best overall trade-off between performance and optimization stability. We therefore use c=1.
> c|VCM I2V (R1/mAP)|VCM V2I (R1/mAP)|BUPT I2V (R1/mAP)|BUPT V2I (R1/mAP)
> -|-|-|-|-
> 0.5|72.2/59.0|72.4/55.9|60.7/57.7|59.2/54.3
> 1.0|76.0/63.2|77.5/60.8|67.6/64.6|68.6/64.2
> 1.5|75.5/61.3|77.6/59.4|66.5/63.9|69.0/65.4
> 2.0|74.7/59.5|76.0/57.4|66.3/63.0|67.4/64.0
>
> **[Q4]** Thank you for pointing out that the scalar case of $\otimes_c$ was not clearly defined. The scalar and matrix versions follow the same principle: both apply the corresponding Euclidean operation in the tangent space at the origin and then map the result back to the manifold. Specifically, the intended scalar Möbius multiplication:
> $$r \otimes_c x=\frac{1}{\sqrt{c}}
> \tanh\Big(r\tanh^{-1}(\sqrt{c}\||x\||)\Big)\frac{x}{\||x\||} = \exp_{\bf{o}}^c\big(r\log_{\bf{o}}^c(x)\big),$$
> while the Möbius matrix-vector multiplication in Appendix A should be revised as:
> $$M \otimes_c^M x=\frac{1}{\sqrt{c}}\tanh\Big(
> \frac{\||Mx\||}{\||x\||}\tanh^{-1}(\sqrt{c}\||x\||)
> \Big)\frac{Mx}{\||Mx\||}= \exp_{\bf{o}}^c\big(M\log_{\bf{o}}^c(x)\big).$$
> Accordingly, Eq. (4)/(14) should be interpreted as scalar Möbius multiplication followed by iterative Möbius addition:
> $$\bigoplus_i (\alpha_i \otimes_c v_i).$$
> We will revise Eq.(4) and Eq.(14) accordingly and explicitly define scalar Möbius multiplication in the revision.

---

> > ### Author Rebuttal · Reviewer_GWrv · 2026-04-02
> >
> > Thanks for the detailed response. It would have been encouraging to see this work extend beyond the visible-infrared modality pair, as recent works have started to shift toward multi-modal settings (e.g., text, sketch).
> > Nevertheless, this paper is a solid work, and with that reason I have raised my score.

---

> > > ### Author Response · Authors · 2026-04-02
> > >
> > > Thank you for your thoughtful follow-up and for recognizing our rebuttal. We sincerely appreciate your time, constructive feedback, and positive update.

---

### Official Review · Reviewer_rWqi · 2026-03-10

**Soundness:** 3
**Presentation:** 3
**Significance:** 3
**Originality:** 4
**Overall Recommendation:** 5
**Confidence:** 4

**Summary:**

This paper proposes Hyperbolic Hierarchical Alignment (HHA) for video-based visible-infrared person re-identification. The key idea is to leverage hyperbolic geometry to better capture hierarchical spatio-temporal variations in video tracklets and facilitate cross-modality alignment. The framework integrates a Hyperbolic Hierarchical Spatio-Temporal Aggregator (HHSA) for geometry-aware temporal modeling and a Geometry-Aware Modality Alignment (GMA) module for aligning visible and infrared representations through modality centroids and identity prototypes. The overall approach is conceptually clean and well-motivated, offering a principled geometry-aware formulation that could serve as a strong reference baseline for future research. Experiments on HITSZ-VCM and BUPTCampus demonstrate state-of-the-art performance.

**Compliance With Llm Reviewing Policy:**

Affirmed.

**Final Justification:**

The authors’ rebuttal is convincing and responsive to my main concerns. In particular, they further clarified the advantages of hyperbolic modeling over Euclidean designs in this task, explained the roles of the key components more clearly, and provided additional empirical evidence to support their claims. This further strengthens my confidence in the paper’s technical motivation and method design.

Overall, I believe this is a novel and technically sound work, supported by solid experiments, and I therefore support acceptance.

**Key Questions For Authors:**

1.The paper introduces hyperbolic geometry into the VVI-ReID framework. Could the authors further clarify what key advantages hyperbolic modeling provides compared with conventional Euclidean designs in this task?

2.The proposed framework integrates several components (HHSA, HGI, DGF, and GMA). Could the authors summarize which aspects of the design are most critical for enabling the hyperbolic modeling of spatio-temporal relationships?

**Limitations:**

The main limitations of the paper are mostly related to presentation and analysis rather than the core methodology. For example, the paper lacks qualitative retrieval visualizations and contains a few minor formatting or description issues (e.g., table captions and metric formatting), which could be further improved to enhance clarity.

**Strengths And Weaknesses:**

Strengths：
1.The motivation and design choices are clearly explained, and the method is presented in a detailed and easy-to-follow manner.

2.The proposed method is relatively simple yet well-structured, with both theoretical motivation and practical effectiveness, making it a strong reference baseline for future research in this direction.

3.The paper introduces hyperbolic representation learning into the VVI-ReID setting, providing a novel geometric perspective for modeling cross-modal video representations.

4.The method achieves state-of-the-art results on the evaluated benchmarks, demonstrating the effectiveness of the proposed design. The experimental results also provide empirical evidence that modeling spatio-temporal relationships in hyperbolic space can be beneficial for this task.

Weaknesses:

1.While the paper presents comprehensive quantitative and ablation studies, adding qualitative visualizations (e.g., retrieval results) could further help illustrate the effectiveness of the proposed hyperbolic modeling.

2.There appears to be a minor formatting issue around line 439, where the performance metrics on the left side seem to be missing percentage symbols (e.g., “70.6%/53.8” should likely be “70.6%/53.8%”).

3.Some table captions could be more descriptive. For example, in Table 6, the meanings of some metrics (e.g., Latency) could be explained more clearly in the caption to improve readability.

---

> ### Author Rebuttal · Authors · 2026-03-31
>
> Thank you for your positive and constructive comments. We respond point by point below (W: Weakness, Q: Question).
>
> **[W1]** We thank the reviewer for this helpful suggestion. Qualitative retrieval visualizations are already provided in the supplementary material. In addition, we further provide representative visualizations at [Visual analysis](https://anonymous.4open.science/r/anon-figures-F024), to more directly illustrate the effect of the proposed hyperbolic modeling. We will add a clearer pointer in the main paper to improve visibility.
>
> **[W2,W3]** We thank the reviewer for these helpful comments on presentation. The missing percentage symbol is indeed a typo and will be corrected in the revision. We also agree that some table captions can be more descriptive. In particular, we will revise Table 6 to clarify the reported metrics, e.g., “Efficiency comparison under a per-frame setting. Params, FLOPs, Latency, and FPS denote model size, computation cost, inference time, and throughput, respectively.”
>
> **[Q1]** We thank the reviewer for this insightful question. Compared with conventional Euclidean designs, we believe the key advantages of hyperbolic modeling in VVI-ReID are mainly reflected in two aspects.
>
> **(1) Better modeling of structured spatio-temporal variations.**
> A VVI-ReID tracklet contains diverse temporal variations, such as pose change, viewpoint change, occlusion, and motion blur. Rather than being simple flat perturbations, these variations need to be aggregated into a structured sequence representation. Due to its negative curvature, hyperbolic space exhibits exponential volume growth, which provides more capacity to accommodate this progressively expanding structure with lower distortion and thus benefits temporal aggregation.
>
> Empirically, we validate this in two ways. First, Table 5 shows that the hyperbolic temporal modeling in HHSA consistently outperforms its Euclidean counterpart in the VVI-ReID setting. Second, after removing the cross-modal-specific GMA, HHSA still improves performance on single-modal video ReID benchmarks, indicating that the benefit of hyperbolic temporal modeling is not limited to cross-modal learning.
>
> Single-modal transfer results without GMA.
> |Method|MARS (R1/mAP)|LS-VID (R1/mAP)|
> |-|-|-|
> |DS-VReID[1]|92.3/87.6|88.2/78.7|
> |B + HHSA|92.4/88.8|89.9/82.6|
>
> [1] Leng et al. Dual-space video person re-identification. IJCV, 2025.
>
> **(2) More suitable geometry for cross-modality alignment.**
> Visible and infrared sequences usually exhibit different appearance distributions and variation patterns. Direct alignment in Euclidean space may weaken the intrinsic structure of each modality. In contrast, hyperbolic space provides a geometry-aware way to align cross-modality representations toward shared identity semantics while better preserving their structural differences. This is why HHA extends hyperbolic modeling beyond HHSA and further performs visible-infrared alignment through GMA (HMA + HPA). Consistently, Table 5 shows that HMA/HPA further improve cross-modality retrieval on top of HHSA.
>
> We also provide additional quantitative and visual evidence (see our response to **Reviewer GWrv, [W2]**), showing that HHA exhibits lower $\delta$-hyperbolicity and clearer prototype-centered organization than its Euclidean counterpart.
>
> **[Q2]** Thank you for this question. We view HGI as the core of hyperbolic spatio-temporal modeling, DGF as the key enhancement module for fusing complementary cues, and GMA as the module that further strengthens cross-modality alignment.
>
> **(1) HGI is the core of hyperbolic spatio-temporal modeling.**
> Within HHSA, HGI is the core component because it performs frame-level interaction directly in hyperbolic space and organizes temporal cues into more structured representations. Table 5 supports this point: when temporal interaction remains Euclidean, the gain is limited, whereas hyperbolic interaction brings clearer and more consistent improvements.
>
> **(2) DGF complements HGI by fusing complementary cues.**
> After HGI captures hyperbolic temporal relations, DGF further fuses the Euclidean and hyperbolic readouts, combining complementary appearance and structural cues. Its role is to enhance the discriminability and robustness of the representation built by HGI, rather than to serve as the core source of hyperbolic temporal modeling.
>
> **(3) GMA further strengthens cross-modality alignment.**
> GMA builds on HHSA by aligning visible and infrared representations on the hyperbolic manifold through HMA and HPA. Its main role is to translate the learned hyperbolic structure into stronger cross-modality retrieval performance, rather than to serve as the primary source of temporal modeling.
>
> In summary, HGI is the core, DGF is the key enhancement module, and GMA further improves cross-modality alignment; together, they form a unified hyperbolic VVI-ReID framework.

---

> > ### Author Rebuttal · Reviewer_rWqi · 2026-04-02
> >
> > After reviewing all the comments and rebuttals, all the concerns have been overcome. I will keep my positive score.

---

> > > ### Author Response · Authors · 2026-04-02
> > >
> > > Thank you very much for the positive acknowledgement. We sincerely appreciate your time, careful reading, and supportive feedback.

---

### Official Review · Reviewer_8f8T · 2026-03-11

**Soundness:** 3
**Presentation:** 4
**Significance:** 3
**Originality:** 3
**Overall Recommendation:** 4
**Confidence:** 2

**Summary:**

This paper introduces Hyperbolic Hierarchical Alignment (HHA), a new approach for video-based visible-infrared person re-identification (VVI-ReID). The core motivation stems from the limitations of standard Euclidean spaces, which struggle to capture the underlying hierarchical nature of cross-modal temporal dynamics, often resulting in cue crowding and distorted feature representations. To overcome this, the authors map video tracklets into a hyperbolic space using a Poincaré ball. Architecture-wise, the framework relies on two main components: a spatio-temporal aggregator (HHSA) that fuses temporal information via Hyperbolic Geometry Interaction and Dual-Geometry Fusion, and a modality alignment module (GMA) that minimizes the visible-infrared domain gap through hyperbolic prototype alignment. Empirical results demonstrate that the proposed method achieves state-of-the-art accuracy on the HITSZ-VCM and BUPTCampus datasets.

**Compliance With Llm Reviewing Policy:**

Affirmed.

**Final Justification:**

My final recommendation is a Weak Accept, upgraded from my initial Weak Reject.

The paper presents a highly original and theoretically sound approach by applying hyperbolic geometry to resolve the cue crowding problem in VVI ReID. My initial concerns regarding empirical soundness and reproducibility were fully resolved during the rebuttal. The authors provided extensive 8 seed experiments to prove statistical stability and supplied clear pseudocode.

The absolute performance margins over strong baselines remain relatively modest, so I cannot give a higher score. However, the method is now proven to be technically solid and robust. The strong originality and clear presentation outweigh this weakness and make it a valuable contribution.

**Key Questions For Authors:**

1. Given the narrow margins against strong baselines (e.g., X-ReID on BUPTCampus ), could you provide the mean and standard deviation over 3 to 5 random seeds for the main results in Table 1 and Table 2?
2. How was the numerical instability inherent in Möbius operations and manifold projection maps mitigated during training? To ensure the reproducibility of the results, can you provide the source code or include detailed pseudo-code for the HHSA and GMA modules, particularly regarding the implementation of Hyperbolic Geometry Interaction (HGI) and Dual-Geometry Fusion (DGF)?
3. Please explicitly clarify the finetuning strategy for the CLIP ViT-B-16 backbone. Which specific layers were frozen, and which were updated?
4. Is the proposed HHSA strictly coupled with the Vision Transformer architecture, or can these hyperbolic interactions be seamlessly integrated into CNN-based backbones?

**Limitations:**

yes

**Strengths And Weaknesses:**

Strengths:
- The application of hyperbolic geometry to address cue crowding in VVI-ReID is highly original and theoretically well-motivated. The authors clearly demonstrate why the exponential volume expansion of the Poincaré ball is better suited for the hierarchical nature of video tracklets compared to the polynomial expansion of Euclidean space.

- The design of Geometry-Aware Modality Alignment (GMA) effectively establishes a geometry-consistent embedding space by coupling modality centroids (HMA) and anchoring them to shared identity prototypes (HPA) on the manifold.

Weaknesses:
- The performance margins over recent strong baselines (e.g., VLD, X-ReID) are narrow and sometimes inconsistent. For instance, on BUPTCampus under the V2I protocol (sequence length 6), the proposed method achieves 64.2% mAP compared to VLD’s 63.0%. More critically, at sequence length 10, HHA (65.7% mAP) actually underperforms X-ReID (65.9% mAP). This directly contradicts the authors' assertion that hyperbolic spatio-temporal modeling naturally excels in more complex and diverse data scenarios.
- Given these marginal improvements and instances of underperformance, relying on single-seed point estimates is highly inadequate to establish robustness. The complete absence of multi-seed experiments and variance reporting makes it impossible to determine whether the claimed state-of-the-art results are statistically significant or merely the result of random seed variation.
- The implementation details specify the use of a "CLIP backbone with a ViT-B-16 vision encoder", but the paper fails to document the exact finetuning protocol. It is completely unclear whether the vision encoder is fully finetuned, partially frozen, or if the text encoder of CLIP was utilized during initialization or training. This ambiguity further degrades reproducibility.

---

> ### Author Rebuttal · Authors · 2026-03-31
>
> Thank you for your careful review and constructive comments. We respond point by point below (W:Weakness,Q:Question).
>
> **[W1]** Thank you. We agree that the original wording was too strong. Our intended claim is not that HHA uniformly outperforms prior methods in every setting, but that it offers an effective geometric formulation for modeling complex temporal variations.
> On BUPTCampus, HHA achieves 64.2% mAP under the 6-frame setting, surpassing VLD (63.0%). Under the 10-frame setting, it achieves 65.7% mAP, comparable to X-ReID (65.9%), while obtaining higher Rank-1 accuracy (69.9% vs 68.8%). Therefore, HHA achieves competitive performance with a relatively simple design, rather than consistently outperforming strong baselines across settings.
>
> **[W2,Q1]** Following your suggestion, we report 8-seed mean ± std results for main BUPT comparisons and ablations on BUPT and VCM. The improvements remain stable across runs, indicating that the gains are not due to seed variance.
>
> Main results(mean±std). V6/B10:VCM 6 frames/BUPT 10 frames.
> Set|M|I2V R1|mAP|V2I R1|mAP
> -|-|-|-|-|-
> V6|VLD|74.3|60.2|74.6|58.6
> V6|HHA|75.6±1.7|62.6±1.6|76.3±0.9|59.9±1.1
> B10|X-ReID|68.2|68.5|68.8|65.9
> B10|HHA|70.4±0.8|67.5±0.6|70.4±1.5|65.7±0.8
>
> Ablation on BUPT (6 frames).
> B|HHSA|HMA|HPA|I2V R1|I2V mAP|V2I R1|V2I mAP
> -|-|-|-|-|-|-|-
> ✓|✗|✗|✗|40.8|42.5|40.8|43.6
> ✓|✓|✗|✗|63.9±1.4|62.6±1.3|63.8±1.6|61.3±2.1
> ✓|✓|✓|✗|66.2±1.6|64.1±1.0|65.6±1.6|62.2±1.4
> ✓|✓|✓|✓|67.4±1.7|65.5±1.0|68.0±1.5|64.2±1.3
>
> Ablation on VCM (6 frames).
> B|HHSA|HMA|HPA|I2V R1|I2V mAP|V2I R1|V2I mAP
> -|-|-|-|-|-|-|-
> ✓|✗|✗|✗|64.4|49.8|63.6|47.7
> ✓|✓|✗|✗|71.1±0.9|57.9±1.0|73.2±0.9|55.9±0.7
> ✓|✓|✓|✗|73.9±1.2|59.1±0.9|75.0±0.9|57.2±0.8
> ✓|✓|✓|✓|75.6±1.7|62.6±1.6|76.3±0.9|59.9±1.1
>
> **[W3,Q3]** Thank you for pointing out this ambiguity. We use the pretrained CLIP ViT-B/16 visual encoder for initialization and fine-tune the entire visual backbone end-to-end; no visual layers are frozen. The CLIP text encoder is not used during training or inference, and HHA does not include any text-guided branch or prompt learning.
>
> **[Q2]** Numerical instability in hyperbolic space mainly arises when Euclidean operations are mixed with manifold features or when points approach the ball boundary, where functions such as $artanh(\cdot)$ may become undefined. To avoid this, we keep the computation geometry-consistent and apply explicit clipping/projection safeguards so that all features remain strictly inside the valid ball region. Prototypes are optimized in tangent space and mapped to the manifold only in forward pass. We will clarify this in the revision.
>
> For reproducibility, we provide pseudo-code, and core implementation in image form via **https://anonymous.4open.science/r/anon-figures-F024**. Code will be publicly available.
>
> ```
> class HHSA:
>     def __init__(self):
>         self.hgi_block = HyperbolicBlock()
>         self.euc_cross_attn = CrossAttention()
>         self.hyp_cross_attn = HyperbolicCrossAttention()
>
>     def forward(self, x_seq):
>         h_memory = []
> //HGI   for layer in backbone:
>             x_seq = layer(x_seq)
>             if is_hgi_layer(layer):
>                 x_cls = extract_cls_token(x_seq)
>                 z_cls = exp_map(x_cls)
>                 z_cls_refined = self.hgi_block(z_cls)
>                 x_cls_refined = log_map(z_cls_refined)
>                 h_memory.append(x_cls_refined)
>                 x_seq = update_cls_token(x_seq,x_cls_refined)
>
> //DGF   x_q = extract_cls_token(x_seq)
>         x_euc = self.euc_cross_attn(query=x_q,memory=h_memory)
>         z_q = exp_map(x_q)
>         z_memory = exp_map(h_memory)
>         z_hyp = self.hyp_cross_attn(query=z_q,memory=z_memory)
>         x_hyp = log_map(z_hyp)
>         x_final = layer_norm(x_euc+x_hyp)
>         return x_final
>
> class GMA:
>     def __init__(self,num_classes,in_dim):
>         self.proto_tan = Parameter(init_random_normal(num_classes, in_dim) * 1e-2)
>         self.hyp_classifier = HyperbolicClassifier(self.proto_tan)
>
>     def compute_loss(self, x_hyp, pids, modal):
>         x_tan = log_map(x_hyp),L_hpa=0,L_hma=0
>         hyp_logits = self.hyp_classifier(x_hyp)
>         L_hce = CE(hyp_logits,pids)
>         for y in unique(pids):
>             vis_tan = get_samples(x_tan,y,modal=='VIS')
>             ir_tan = get_samples(x_tan,y,modal=='IR')
>             mu_vis = exp_map(mean(vis_tan))
>             mu_ir = exp_map(mean(ir_tan))
>             p_y = exp_map(self.proto_tan[y])
>             L_hpa = L_hpa + hyp_sqdist(mu_vis,p_y) + hyp_sqdist(mu_ir,p_y)
>             L_hma = L_hma + hyp_sqdist(mu_vis,mu_ir)
>         L_hpa = L_hpa + L_hce
>         return L_hpa,L_hma
> ```
>
> **[Q4]** HHSA is instantiated on a ViT, but its formulation is not inherently tied to ViT-specific token mechanics. The key requirement is a sequence of frame-level representations, which could also be obtained from a CNN and then fed into HGI and DGF for sequence modeling. We only validate HHSA on ViT here, but the formulation is not restricted to it.

---

> > ### Author Rebuttal · Reviewer_8f8T · 2026-04-02
> >
> > I thank the authors for the clear and constructive rebuttal. While the absolute performance margins over the baselines remain somewhat modest, the provided 8-seed experiments and pseudo-code successfully resolve my core concerns regarding statistical stability and reproducibility. Therefore, I consider this a solid work and have raised my score.

---

> > > ### Author Response · Authors · 2026-04-02
> > >
> > > Thank you for your constructive feedback and positive update. We are glad our rebuttal addressed your concerns. The code will be made publicly available.

---

### Decision · Program_Chairs · 2026-04-30

**Decision:**

Accept (regular)

**Comment:**

This paper introduces Hyperbolic Hierarchical Alignment (HHA), a new approach for video-based visible-infrared person re-identification (VVI-ReID). It integrates a Hyperbolic Hierarchical Spatio-Temporal Aggregator (HHSA) for geometry-aware temporal modeling and a Geometry-Aware Modality Alignment (GMA) module for aligning visible and infrared representations through modality centroids and identity prototypes.

During the review process, several key concerns were raised, mainly regarding the implementation details, instances of underperformance, the justification of the hyperbolic feature space, and the ablation study. After the rebuttal, all reviewers acknowledged that their concerns had been fully resolved.

Overall, this work meets the requirements of the conference, but the authors should revise the manuscript to address the remaining issue, especially the clarification of the justification. For these reasons, I recommend acceptance.